# Combating acquired resistance to MAPK inhibitors in melanoma by targeting Abl1/2-mediated reactivation of MEK/ERK/MYC signaling

Rakshamani Tripathi [1], Zulong Liu[1], Aditi Jain[1,7], Anastasia Lyon[1], Christina Meeks[1], Dana Richards[2], Jinpeng Liu[3], Daheng He[3], Chi Wang[3], Marika Nespi[4], Andrey Rymar[4], Peng Wang[5], Melissa Wilson [6] & Rina Plattner [1✉]

Metastatic melanoma remains an incurable disease for many patients due to the limited success of targeted and immunotherapies. BRAF and MEK inhibitors reduce metastatic burden for patients with melanomas harboring *BRAF* mutations; however, most eventually relapse due to acquired resistance. Here, we demonstrate that ABL1/2 kinase activities and/or expression are potentiated in cell lines and patient samples following resistance, and ABL1/2 drive BRAF and BRAF/MEK inhibitor resistance by inducing reactivation of MEK/ERK/MYC signaling. Silencing/inhibiting ABL1/2 blocks pathway reactivation, and resensitizes resistant cells to BRAF/MEK inhibitors, whereas expression of constitutively active ABL1/2 is sufficient to promote resistance. Significantly, nilotinib (2nd generation ABL1/2 inhibitor) reverses resistance, in vivo, causing prolonged regression of resistant tumors, and also, prevents BRAFi/MEKi resistance from developing in the first place. These data indicate that repurposing the FDA-approved leukemia drug, nilotinib, may be effective for prolonging survival for patients harboring BRAF-mutant melanomas.

---

[1] Department of Pharmacology and Nutritional Sciences, University of Kentucky College of Medicine, Lexington, KY 40536, USA. [2] Department of Pathology, University of Kentucky College of Medicine, Lexington, KY 40536, USA. [3] Biostatistics and Bioinformatics Shared Resource Facility, Markey Cancer Center, University of Kentucky College of Medicine, Lexington, KY 40536, USA. [4] Plexxikon Inc., Berkeley, CA 94710, USA. [5] Department of Internal Medicine, University of Kentucky, College of Medicine, Lexington, KY, USA. [6] Department of Medical Oncology, Sidney Kimmel Cancer Center, Thomas Jefferson University, Philadelphia, PA, USA. [7] Present address: The Jefferson Pancreas, Biliary and Related Cancer Center, Department of Surgery, Sidney Kimmel Cancer Center, Thomas Jefferson University, Philadelphia, PA, USA. ✉email: rplat2@uky.edu

Melanoma incidences continue to increase, and the disease remains incurable for many due to its metastatic nature and high rate of therapeutic resistance (Cancer. Net). Immunotherapy increases the cure rate for some advanced cases; however, 50–60% of patients do not respond and some cannot tolerate the therapy[1,2]. Moreover, for those with high metastatic burden, immunotherapy may not sufficiently reduce tumor burden or induce remission[1,2]. Melanomas harboring activating BRAF-V600E/K mutations (70–88%) can be treated with BRAF inhibitors (BRAFi; e.g. vemurafenib, dabrafenib)[3]; however, some BRAF-mutant melanomas are intrinsically resistant, and many that respond initially, develop resistance during treatment (median progression-free survival—PFS, 5–7 months)[4]. Since a common BRAFi resistance mechanism involves BRAF-independent activation of MEK, combining BRAF and MEK inhibitors (MEKi; e.g. trametinib, cobimetinib) delays resistance (median PFS, 12 months)[3]. However, the vast majority still succumb to the disease due to acquired resistance. Tumors circumvent the effects of BRAFi and BRAFi/MEKi via (1) reactivation of ERK due to mutations in MAP2K1 (MEK1), MAP2K2 (MEK2)[5], MAP3K8 (COT; TPL-2)[6], NRAS or KRAS[5], overexpression of receptor tyrosine kinases (RTKs)[5], or expression of BRAF-V600E splice variants[3]; (2) overexpression of the master melanocyte transcription factor, MITF[5]; or (3) PI3K/AKT activation by RTKs[7] or as a result of PTEN or AKT mutations[5,7,8]. Thus, new drug combinations are needed to increase the 5-year survival rate for patients with advanced disease.

ABL1 (a.k.a. c-Abl) and ABL2 (a.k.a. Arg; Abl-related-Gene) non-receptor tyrosine kinases are best known for their oncogenic role in human leukemia, where they are partners in chromosomal translocations (e.g. BCR-ABL1, ETV6-ABL2) that activate the kinases and drive disease progression[9]. The first-generation ABL inhibitor, imatinib, has been used for decades to treat BCR-ABL1-driven chronic myelogenous leukemia (CML), and second (e.g. nilotinib) and third (e.g. ponatinib)-generation drugs were developed to combat imatinib resistance[9]. Nilotinib, which is more efficacious than imatinib, is well-tolerated (most common adverse effects are mild liver damage, nausea, vomiting, rash, and musculoskeletal pain)[10].

Accumulating evidence supports a role for ABL1/2 in solid tumors where they are activated downstream of other kinases or are mutationally activated in a small subset[9,11]. ABL1 activation in cytoplasm/plasma membrane compartments has pro-survival/oncogenic roles, whereas nuclear ABL1 often induces apoptosis[9]. In contrast, ABL2, which promotes invasion (and sometimes, proliferation), is only present in the cytoplasm/plasma membrane[9,12]. ABL1/2 are highly active in 40–60% of melanomas, and drive disease progression[13–15]; however, their role in therapeutic resistance has not been explored. Here, we utilize in vitro and in vivo experiments and patient samples to demonstrate that ABL1/2 drive resistance to BRAFi and BRAFi/MEKi. Although resistance mechanisms differ among the lines queried, targeting ABL kinases reverses resistance in all lines tested, indicating that the kinases are critical signaling nodes. Indeed, nilotinib, an ABL1/2 inhibitor that has been FDA-approved for more than a decade, reverses and prevents resistance, in vivo. These data have strong clinical relevance as they indicate that nilotinib may prolong patient survival in both BRAFi/MEKi-refractory and treatment-naïve settings.

## Results

### ABL1/2 are activated following BRAFi or BRAFi/MEKi resistance.

Since resistance is a major impediment to long-term survival for patients with metastatic melanoma, and ABL kinases are activated by ERK pathway components (BRAF, ERK) and activate ERK[16], we tested whether ABL1/2 drive BRAFi or BRAFi/MEKi resistance. All cell lines utilized in subsequent experiments harbor BRAF-V600E mutations. Clonal, BRAFi-resistant (-BR) lines derived from Mel1617 and 451-Lu were developed by Herlyn and colleagues[17] (Mel1617-BR, 451-Lu-BR). We established polyclonal BRAFi-resistant M14 (M14-BR) and BRAFi/MEKi-resistant (M14-BMR, Mel1617-BMR) lines (see "Methods"), since this more accurately mimics resistance, in vivo. All new resistant lines demonstrate high level resistance to PLX4720 (PLX; BRAFi; -BR) or dabrafenib/trametinib (D/T; BRAFi/MEKi; -BMR) (Supplementary Fig. 1a–c). Importantly, ABL1 and/or ABL2 kinase activities are elevated in resistant lines (compared to parental cells; Fig. 1a and Supplementary Fig. 1d). Kinase activities were assessed using highly sensitive and specific in vitro kinase assays, which involve incubating ABL1 or ABL2 immunoprecipitates with their substrate, GST-CRK, and gamma-$^{32}$P-labeled-ATP[18,19]. Consistent with the notion that cytoplasmic retention induces ABL1's transforming function[9], ABL1 cytoplasmic localization (cytoplasmic:nuclear ratio) also is increased in resistant cells (Fig. 1b and Supplementary Fig. 1e). Cytoplasmic retention of ABL1 is mediated by interaction with 14-3-3 proteins, which sequester ABL1 in the cytoplasm by binding a phosphorylated threonine residue (ABL1-T735)[20,21]. BRAFi/MEKi-resistant lines have increased T735 phosphorylation (Fig. 1c), consistent with fractionation data demonstrating increased ABL1 cytoplasmic retention. Increased ABL1 mRNA in primary melanomas correlates with activated ABL1 signaling and disease progression[14,15]. Importantly, ABL1 mRNA levels also are significantly elevated in patient samples following relapse on BRAFi or BRAFi/MEKi compared to matched samples obtained prior to treatment (using four RNAseq or microarray databases; Fig. 1d and Supplementary Fig. 1f). Taken together, these data demonstrate that ABL1/2 expression and/or activities are potentiated following BRAFi or BRAFi/MEKi resistance.

To identify the mechanism by which ABL1/2 are activated following resistance, we assessed the activity of SRC family kinases (SFKs), which phosphorylate and activate ABL1/2[16,22,23]. SFK activity (pY416) is elevated in many resistant lines, and inhibition of SFK activity using SFK inhibitor, SU6656, prevents ABL1 and ABL2 activation (kinase assays) in M14-resistant lines (Fig. 1e), indicating that SRC activation contributes to ABL1/2 activation during resistance in these cells. Consistent with the kinase assay data, phosphorylation of CRKL (ABL1/2 substrate) on ABL1/2 phosphorylation sites, a well-accepted read-out of ABL1/2 activities[13,24], also is inhibited by SU6656 treatment (Fig. 1e). We used the lowest dose of SU6656 that efficiently reduces SFK activity in melanoma cells (10 μM), and we previously showed that this dose has no direct effect on ABL1 or ABL2 as it does not reduce their activities in cells that lack SFK expression[25]. Interestingly, SU6656 has little effect on ABL1/2 activities in Mel1617-BR; reduces ABL1 but not ABL2 activity in Mel1617-BMR; and reduces ABL2 but not ABL1 activity in 451-Lu-BR (Supplementary Fig. 1g). Thus, although SFKs contribute to ABL1/2 activation in M14-resistant lines, additional kinases also are involved in activating ABL1/2 in Mel1617 and 451-Lu resistant lines.

### Targeting ABL1/2 reverses BRAFi and BRAFi/MEKi resistance.

Next, we tested whether ABL1/2 activation is required for resistance. Parental and BRAFi-resistant lines (-BR) were treated with vehicle, nilotinib, PLX4720 (PLX; BRAFi), or the combination. As expected, parental cells are highly sensitive to PLX alone (cell viability, colony-forming assays), whereas resistant cells are impervious to the effects of the drug (Fig. 2a–e and Supplementary Fig. 2a, b). Treatment with nilotinib alone has little effect

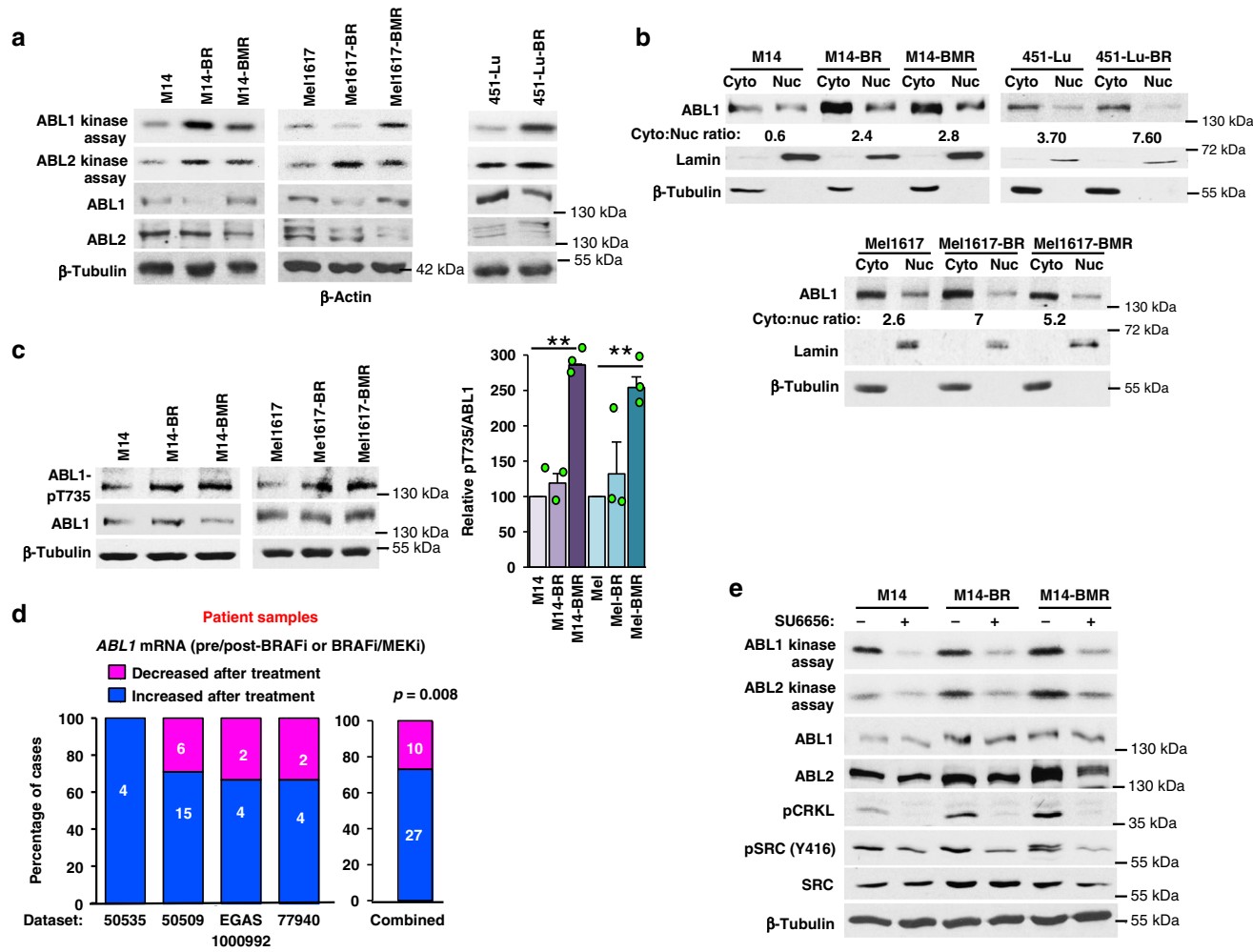

**Fig. 1 ABL kinase activities are potentiated following resistance to BRAF and BRAF/MEK inhibitors. a** Parental (M14, Mel1617, 451-Lu), BRAFi (-BR), or BRAFi/MEKi-resistant (-BMR) cell lines were serum-starved and subjected to in vitro kinase assay (top two panels) or western blot (rest of the panels) for total protein expression. For kinase assays, ABL1 was immunoprecipitated with K12 antibody, whereas ABL2 was immunoprecipitated with ABL2-specific antibody[19,62], and IPs incubated with substrate (GST-CRK) and radiolabeled (gamma-$^{32}$P)-ATP (see "Methods"). Kinase quantitation for $n = 3–5$ biological replicates is shown in Supplementary Fig. 1d. **b** Resistant cell lines (**a**) were fractionated into nuclear (nuc) and cytoplasmic (cyto) fractions, equal percentages of cytoplasmic/nuclear fractions loaded for each line, and fractions blotted with the indicated antibodies. Lamin and tubulin blots were used to demonstrate the purity of nuclear and cytoplasmic fractions, respectively. ABL1 cytoplasmic:nuclear ratio is noted for each cell line, and quantitation for all lines is shown in Supplementary Fig. 1e. A representative of $n = 2$ independent experiments is shown. **c** Lysates from **a** were blotted with the indicated antibodies and quantified. Graph shown is mean ± SEM for three independent experiments. **$p = 0.006$ using single sample $t$-tests (two-sided) and Holm's adjustment for multiple comparisons. **d** Primary human paired melanoma datasets (RNAseq-77940, 50535, 65185, EGAS00001000992; microarray-50509) were analyzed for *ABL1* mRNA levels pre-/post-BRAFi, MEKi, or BRAFi/MEKi treatment. Numbers in each column indicate number of cases. All cases were included; repeated analyses using particular cut-off values are shown in Supplementary Fig. 1f). $p = 0.008$ for the combined set using a two-sided binomial (right). **e** Parental and resistant cells were serum-starved, treated with vehicle (DMSO) or SFK inhibitor, SU6656 (10 μM), for 16 h and lysates were subjected to in vitro kinase assay (top two panels) and western blot (all other panels) as in **a**.

on survival and colony formation in parental or resistant Mel1617 and M14 lines; however, nilotinib modestly reverses intrinsic resistance (parental lines; viability assays) and reverses acquired resistance (resistant lines) when combined with PLX (Fig. 2a–d and Supplementary Fig. 2a, b). Unlike Mel1617-BR and M14-BR, nilotinib alone efficiently reduces the viability and colony formation of 451-Lu-BR, whereas the effects are more modest in the parental line (Fig. 2e, f). Addition of PLX reduces the anti-proliferative/survival effects of nilotinib, although the effects remain greater than PLX alone. Importantly, nilotinib alone (451-Lu-BR) and nilotinib+PLX (Mel1617-BR, M14-BR) induce PARP and caspase-3 cleavage (Fig. 2g), and inhibit colony formation following drug removal (Fig. 2b, d, f), indicating that the effects are

permanent. Similar results are observed with ponatinib, a third-generation ABL1/2 inhibitor with a different set of non-ABL targets as well as with GNF-5, a highly specific, but less potent allosteric ABL1/2 inhibitor with no other known targets (Supplementary Fig. 2c–f)[12]. Moreover, expression of nilotinib-resistant forms of *ABL1/2* (T315I) rescues nilotinib-mediated inhibition of viability in the presence of PLX, indicating that nilotinib's effects are mediated predominantly by ABL1/2 (Supplementary Fig. 2g). Consistent with these data, expression of constitutively active forms of ABL1/2 (P242E/P249E; PP)[23,26] into melanoma cells lacking highly active ABL1/2 and harboring *BRAF-V600E* (WM164; gain-of-function experiment)[16] is sufficient to induce BRAFi resistance (Fig. 2h and Supplementary

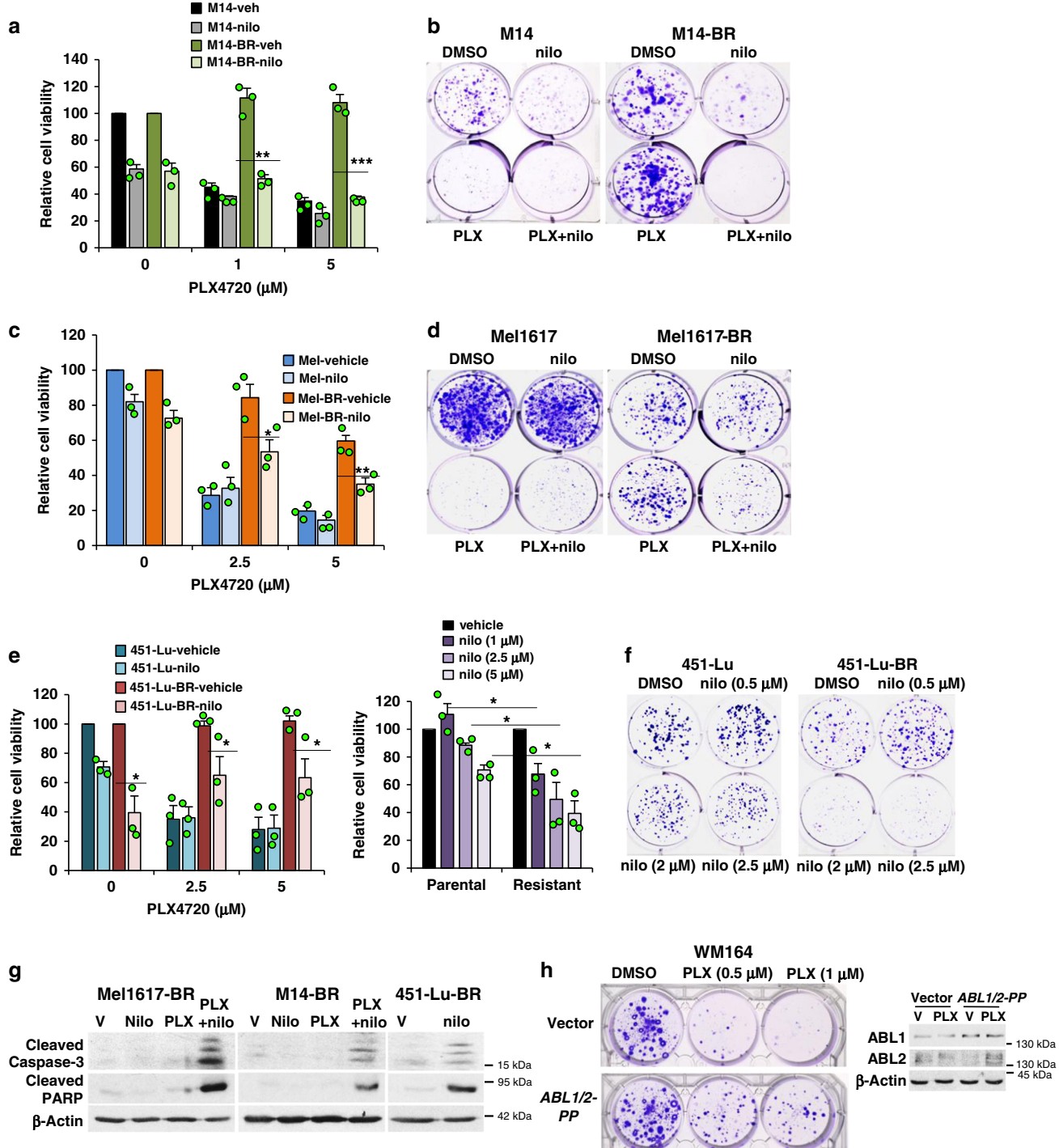

**Fig. 2 ABL1/2 inhibition reverses resistance to BRAF inhibitors. a**, **c**, **e** Viability (CellTiter Glo) assays for cells treated with vehicle (veh) or nilotinib (nilo) alone (**e** (right)) in the absence/presence of BRAFi, PLX4720 (doses shown) for 72 h (**a**, **c**, **e** (left)). Nilotinib doses: **a** 5 or 6 μM; **c**, **e** 5 μM. Results are mean ± SEM for three independent experiments performed in triplicate. Additional drug doses are shown in Supplementary Fig. 2a, b. *$p < 0.05$, **$p \leq 0.01$, ***$p < 0.001$ using two-sample $t$-tests (two sided). Actual $p$ values (left→right): **a** 0.0029, 0.00016; **c** 0.039, 0.0068; **e** (left) 0.016, 0.043, 0.015; **e** (right) 0.019, 0.035, 0.035. **b**, **d**, **f** Colony assays. Cells were treated with nilotinib (**b** 4 μM; **d** 3 μM; **f** doses shown) and/or PLX4720 (2.5 μM) for 7 days, wells washed, plates incubated additional days without drugs (451-Lu/M14-5d; Mel1617-6d) until colonies were well visualized, at which time they were stained with crystal violet. **g** Cells were treated with the indicated drugs for 96 h and detached and attached cells lysed and subjected to western blotting. Nilotinib: Mel1617-BR, 4 μM; M14-BR, 451-Lu-BR, 5 μM. PLX: Mel1617-BR, 4 μM, M14-BR, 3 μM. Representative blots from $n = 2$ independent experiments are shown. V = vehicle. **h** Low-invasive WM164 cells, which lack endogenous activated ABL1/2 and harbor *BRAF-V600E*, were engineered to stably express constitutively active forms of *ABL1* and *ABL2* (PP) or vectors. Cells were plated, treated with vehicle or the BRAFi, PLX4720, for 7 days, wells washed, incubated with media without drug for 9 days, stained with crystal violet, and colonies manually counted. Levels of ABL1/2 proteins are shown on the right. Cells were maintained in the presence of PLX but removed from drug two days prior to plating for experiments. Colony assay shown is representative of $n = 4$ (0.5 μM) or $n = 2$ (1 μM) independent experiments (see Supplementary Fig. 2h). V = vehicle = DMSO.

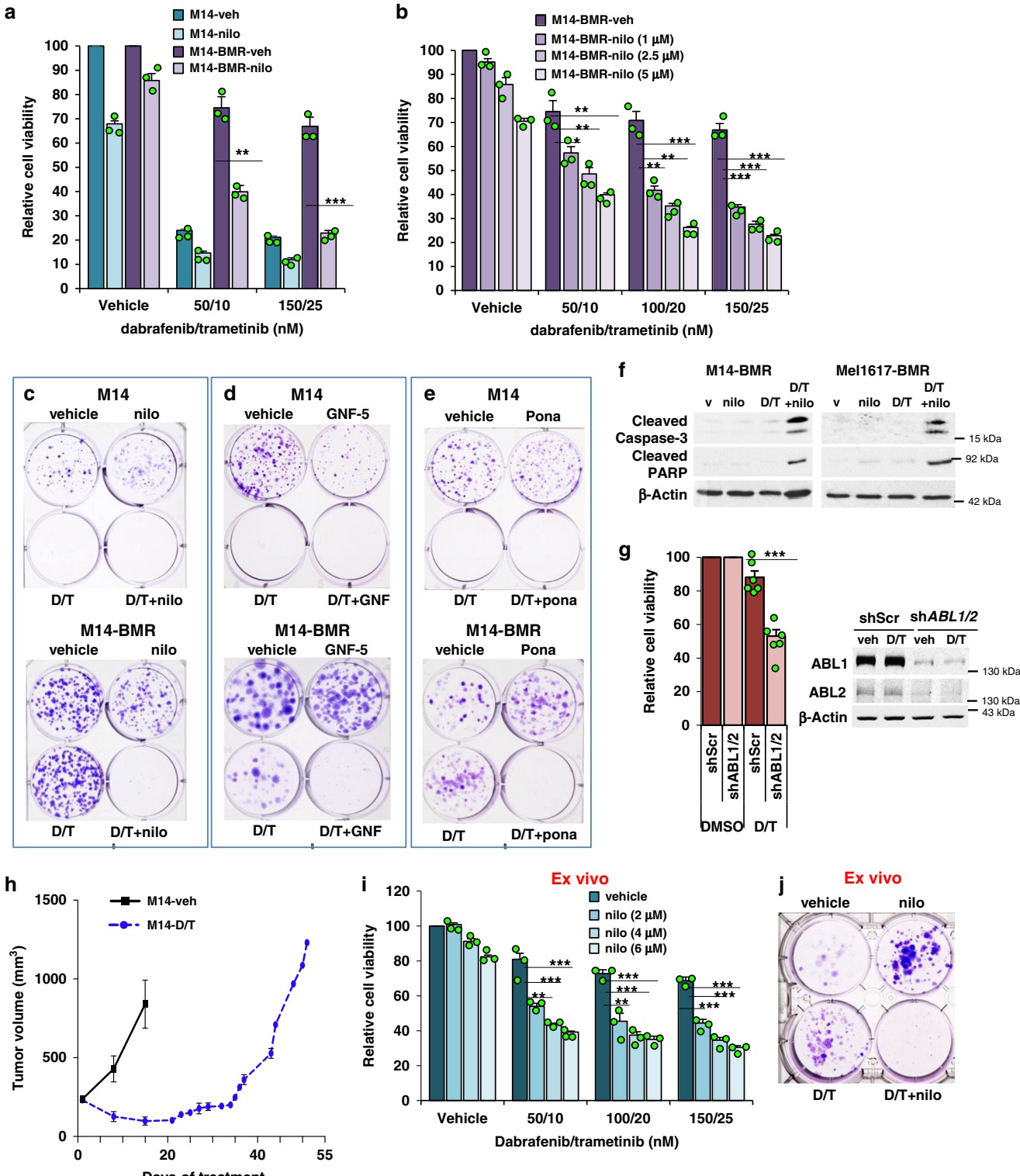

Fig. 2h). Thus, ABL kinases drive acquired BRAFi resistance, and ABL1/2 inhibitors cooperate with BRAFi to prevent viability in some lines (M14-BR, Mel1617-BR) or are highly effective on their own for others (451-Lu-BR).

Metastatic melanoma is currently treated with BRAFi/MEKi since it increases PFS over BRAFi alone[4]. Thus, we examined whether ABL1/2 also drive resistance to BRAFi/MEKi. Importantly, targeting ABL1/2 (nilotinib, ponatinib, GNF-5, *ABL1/2* shRNA) in BRAFi/MEKi-resistant lines (M14-BMR, Mel1617-

BMR) reverses resistance to dabrafenib+trametinib (D/T; BRAFi/MEKi) by inhibiting viability, colony formation, and inducing PARP and caspase-3 cleavage in the presence of D/T (Fig. 3a–g and Supplementary Fig. 3a–c). Moreover, re-expression of *ABL1/2* rescues *ABL1/2* shRNA-mediated inhibition of survival (Supplementary Fig. 3d). To mimic clinically relevant mechanisms of resistance, we also developed a cell line from melanoma cells that developed BRAFi/MEKi resistance, in vivo (Fig. 3h). Importantly, nilotinib also reverses D/T

**Fig. 3 Blocking ABL1/2 activity reverses resistance to BRAF + MEK inhibitors. a, b** CellTiter Glo viability assays using parental and BRAF/MEK-inhibitor resistant M14 cells treated with nilotinib (nilo; 2.5 μM) in the absence/presence of BRAFi/MEKi (dabrafenib/trametinib; D/T) for 72 h. Results are mean ± SEM for three independent experiments performed in triplicate. Actual p values left→right: **a** 0.0015, <0.0001 using two-sided, two-sample t-tests; **b** 50/ 10: 0.031, 0.008, 0.0018; 100/20: 0.0022, 0.0085, 0.00031; 150/25: 0.00037, 0.00018, <0.0001 using two-sample t-tests (two sided). **c–e** Colony assays. Cells were treated with vehicle, D/T (100 nM/20 nM) in the absence or presence of nilotinib (2.5 μM), GNF-5 (GNF; 10 μM), or ponatinib (pona; 100 nM) for 7 days, washed, and incubated in the absence of drugs for an additional 6 days (**c**, **e**) or treated for 13 days (**d**). **f** Cells were treated with nilotinib (M14-BMR-5 μM; Mel1617-BMR-4 μM) in the absence or presence of D/T (150 nM/25 nM) for 96 h, and detached and attached cells lysed and subjected to western blotting. Representative blots from n = 2 independent experiments are shown. V = vehicle. **g** M14-BMR cells expressing scrambled (shScr) or IPTG-inducible shRNA targeting ABL1 and ABL2 (shABL1/2) were treated with IPTG (1 mM) for 5 days prior to plating, treated with vehicle or D/ T (150 nM/25 nM) for 72 h, and cell viability assessed with CellTiter Glo. Results are mean ± SEM for two vector and two shRNA clones and three independent experiments for each, expressed as a percentage of vehicle (DMSO). ***p = 7.5e−5 using a two-sample t-test (two sided). Veh = vehicle. Knockdown efficiency (western blots) is shown on the right. **h** M14 BRAFi/MEKi in vivo-resistant cells were obtained by treating mice containing M14 xenografts (200 mm³) with vehicle or D/T and establishing a cell line from resistant tumor tissue on day 49. Graph is mean ± SEM, n = 5 mice/group. **i, j** The established line from **h** was incubated with nilotinib (2.5 μM if doses are not indicated) in the absence/presence of D/T (100 nM/20 nM, if not indicated), and viability assessed with CellTiter Glo assay (mean ± SEM for n = 3 independent experiments performed in triplicate, **i**); and colony formation assessed as above (growth/treatment for 17days; **j**) *p < 0.05, **p ≤ 0.01. ***p < 0.001 using two-sided, one-sample t-tests. Actual p values (left→right): 0.0023, 0.00097, 0.0003, 0.0057, 0.00026, 0.00019, 0.00012, 0.00019, and <0.0001. For all experiments, BRAFi/MEKi-resistant lines were maintained in D/T (100 nM/20 nM) but were cultured for 2 days without D/T prior to plating for experiments.

resistance in this model (Fig. 3i, j), indicating our data have high clinical relevance.

**ABL1/2 drive ERK pathway reactivation during BRAFi resistance.** To uncover the mechanism by which ABL inhibitors reverse resistance to BRAFi, we assessed activation of ERK-dependent signaling pathways since ERK reactivation is a common resistance mechanism. ERK activation results in phosphorylation (pFRA1) and induction of the transcription factors, FRA1/FOSL1 and MYC, which are critical for melanoma development, progression, and resistance[27,28]. In M14 and Mel1617 BRAFi-resistant cells (-BR), treatment with nilotinib or GNF-5 (highly selective but less potent allosteric inhibitor)[12,29] alone has little effect or modestly reduces expression/phosphorylation of signaling proteins in resistant cells (Fig. 4a, b and Supplementary Fig. 4a). Phosphorylation of the ABL1/2 substrate, CRKL, a well-accepted read-out of ABL1/2 activities[13,24], demonstrates the efficiency of nilotinib/GNF-5 inhibition (Fig. 4a–c and Supplementary Fig. 4a). PLX (BRAFi) treatment efficiently inhibits ERK signaling (pERK, pFRA1, MYC) in parental/sensitive lines, whereas it is inefficient at inhibiting these proteins in resistant cells (Fig. 4a, b). Significantly, addition of nilotinib or GNF-5 to PLX blocks reactivation of ERK signaling in resistant cells and also reduces pMEK (S217/S221; activating residues; Fig. 4a, b and Supplementary Fig. 4a). Moreover, gain-of-function experiments demonstrate that ABL1/2 activation is not only necessary but also sufficient to induce pERK/pFRA/MYC (Supplementary Fig. 4b). Interestingly, in 451-Lu-BR cells, nilotinib, which is highly effective on its own at reducing cell viability (Fig. 2e, f), also is highly effective on its own at inhibiting MYC expression and phosphorylation of MEK1/2, ERK1/2, and FRA1 (Fig. 4c). Thus, nilotinib alone (451-Lu-BR) or nilotinib in combination with BRAFi (M14-BR, Mel1617-BR) prevents reactivation of MEK/ ERK signaling during resistance.

To gain additional insight into the mechanism by which targeting ABL1/2 reverses BRAFi resistance, we performed next-generation whole-exome sequencing (WES), identifying new acquired mutations that are present in resistant lines but not in parental cells. Mel1617-BR cells have a mutation in *MAP2K2* (MEK2; Q60P) (Table 1 and Supplementary Data File 1), which also was previously identified in MEKi-resistant Mel1617 and 451-Lu cells[30]. Mel1617-BR cells also have a catalytic domain *PTEN* mutation (G143D); however, the allele frequency is low indicating it is only present in a small subpopulation. M14-BR cells harbor an activating mutation in *NRAS* (Q61R), a known

resistance mechanism in patients (Table 1 and Supplementary Data File 1)[31,32], indicating the line is clinically relevant. Contrary to a previous report[33], our WES analysis did not uncover a *MAP2K1* (MEK1; K57N) mutation in 451-Lu-BR cells, and this result was confirmed using Sanger sequencing (Supplementary Data File 2 and Supplementary Fig. 4c). We obtained 451-Lu-BR soon after it was established (2011)[17], and short tandem repeat (STR) analysis in 2018 confirmed its identity. Interestingly, *MAP2K1*-K57N was identified by WES at a very low allele frequency (1 out of 240 reads), and thus may represent a small subpopulation of the original line. Unlike M14-BR and Mel1617-BR, 451-Lu-BR did not have an obvious ERK pathway mutation. However, we did identify a mutation in the D-3 domain of *MAP2K7* (MKK7; P68A), a MAP2K (like MEK1/2) that binds JNK via D-domains, and cooperates with MAP2K4 (MKK4) to activate JNK[34]. There is accumulating evidence for crosstalk (negative and positive) between JNK and ERK pathways; thus, this mutation could potentially induce ERK reactivation (Table 1 and Supplementary Data File 2)[34,35]. In summary, BRAFi-resistant lines acquire mutations that reactivate ERK signaling via different mechanisms; however, targeting ABL1/2 reverses resistance (or is effective on its own) in all lines, indicating that ABL1/2 are key signaling nodes.

**Targeting ABL1/2 prevents ERK or AKT reactivation during BRAFi/MEKi resistance.** Next, we evaluated the mechanism of ABL1/2-driven resistance to BRAFi/MEKi. In M14-BMR cells, ERK1/2 is reactivated in the presence of BRAFi/MEKi (dabrafenib/trametinib; D/T), leading to induction of pFRA1/FRA1 and MYC (Fig. 5a). Paradoxically, D/T treatment induces upregulation of pMEK1/2 (with concurrent downregulation of total MEK) despite the presence of trametinib (MEKi). This phenomenon has been postulated to be due to phosphorylation of MEK1/2 (S217/ S221; reflects activity) by upstream kinases, which reduces the affinity of trametinib to the allosteric regulatory site[36]. Nilotinib and ponatinib as well as *ABL1/2* shRNA resensitize M14-BMR cells to D/T, reducing pMEK1/2, pERK1/2, pFRA1/FRA1, and MYC expression (Fig. 5a, b and Supplementary Fig. 5a). Thus, ABL1/2 inhibition prevents MEK phosphorylation by upstream kinases.

To gain insight into the mechanism by which targeting ABL1/2 reverses ERK-driven BRAFi/MEKi resistance, we performed next-generation WES. We were unable to discern an ERK pathway mutation in M14-BMR cells that could explain reactivation of MEK/ERK signaling (Supplementary Data File 1). However, we

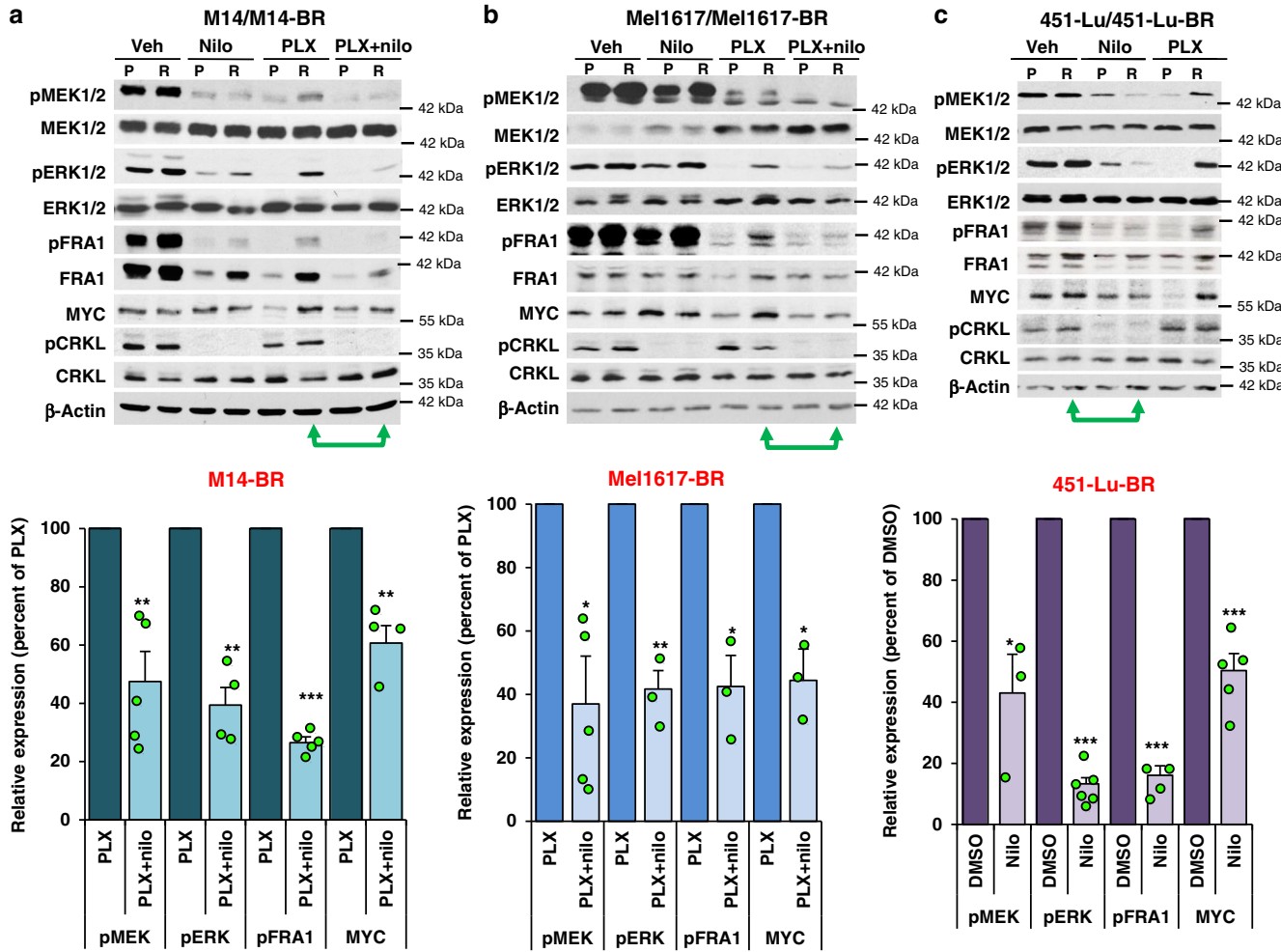

**Fig. 4 Nilotinib blocks reactivation of ERK signaling in BRAFi-resistant cells.** BRAFi (-BR; R) or parental (P) cells were treated with vehicle, nilotinib (nilo; 2.5 μM), PLX4720 (M14 and 451-Lu 1 μM; Mel1617 2.5 μM) or the combination for 24 h, and the resulting lysates probed with the indicated antibodies. Quantitation of key signaling molecules for $n = 3-5$ independent experiments is shown below blots; green arrows indicate lanes quantified. Mean ± SEM is shown. *M14-BR*: pMEK $n = 5$, pERK $n = 4$, pFRA1 $n = 5$, MYC $n = 4$. *Mel1617-BR*: pMEK $n = 5$, pERK $n = 3$, pFRA1 $n = 3$, MYC $n = 3$. *451-Lu-BR*: pMEK $n = 3$, pERK $n = 6$, pFRA1 $n = 4$, MYC $n = 5$. For M14 and Mel1617 cell lines, data are a comparison between PLX + nilotinib vs. PLX lanes, whereas for 451-Lu cell lines, nilotinib alone is compared to vehicle (DMSO). *$p < 0.05$, **$p \leq 0.01$, ***$p < 0.001$ using one-sample t-tests (two-sided). Actual p values (left→right): **a** 0.0069, 0.002, <0.0001, and 0.0072. **b** 0.026, 0.0097, 0.029, and 0.011. **c** 0.046, <0.0001, 0.00011, and 0.0001. pCRKL (substrate of ABL1/2) is an indirect read-out of ABL1/2 activity and indicates the efficiency of nilotinib-mediated inhibition. For all experiments, BRAFi-resistant lines were maintained in PLX (1 μM), but were cultured for 2 days without PLX prior to plating for experiments.

| Table 1 Next-generation whole-exome sequencing. | |
|---|---|
| **Resistant cell lines** | **MAPK pathway mutations** |
| M14-BR | NRAS-Q61R |
| Mel1617-BR | MAP2K1-Q60P |
| 451-Lu-BR | MAP2K7-P68A |
| M14-BMR | MAP4K1-P422Q |

MAPK pathway acquired mutations in resistant lines that are not present in parental lines. See Supplementary Data Files 1 and 2 for complete gene lists.

did identify a *MAP4K1* (MEKKK1; HPK) mutation (P422Q), which was previously identified in a patient with cutaneous melanoma (cBioportal website; Table 1). Although MAP4K1 primarily activates JNK, there is evidence for crosstalk between JNK and ERK pathways[34,35], indicating that this may be a mechanism by which ERK signaling is reactivated.

In Mel1617-BMR cells, modest reactivation of MEK1/2 in the presence of D/T does not result in induction/activation of ERK1/2, FRA1, or MYC, and pMEK1/2 levels are unchanged by combining nilotinib/GNF-5 and D/T (Fig. 5c and Supplementary Fig. 5b). Thus, ABL1/2 activate an ERK-independent resistance pathway in Mel1617-BMR. Importantly, AKT phosphorylation, which is induced by D/T, is reduced by addition of nilotinib or GNF-5 (Fig. 5c and Supplementary Fig. 5b), indicating that nilotinib/GNF also reverse BRAFi/MEKi resistance driven by an ERK-independent mechanism.

**ABL1/2 induce resistance via ERK/MYC reactivation.** The ERK pathway is reactivated during resistance in the majority of lines. To prove that blockade of ERK reactivation is the mechanism by which nilotinib reverses resistance, we engineered M14-BMR cells to express a mutant, gain-of-function (GOF) form of *ERK2* (D231N/R67S)[37]. Significantly, expression of *ERK2*-GOF efficiently reverses nilotinib's anti-survival effects in the presence of D/T, indicating that nilotinib reverses resistance by blocking ERK

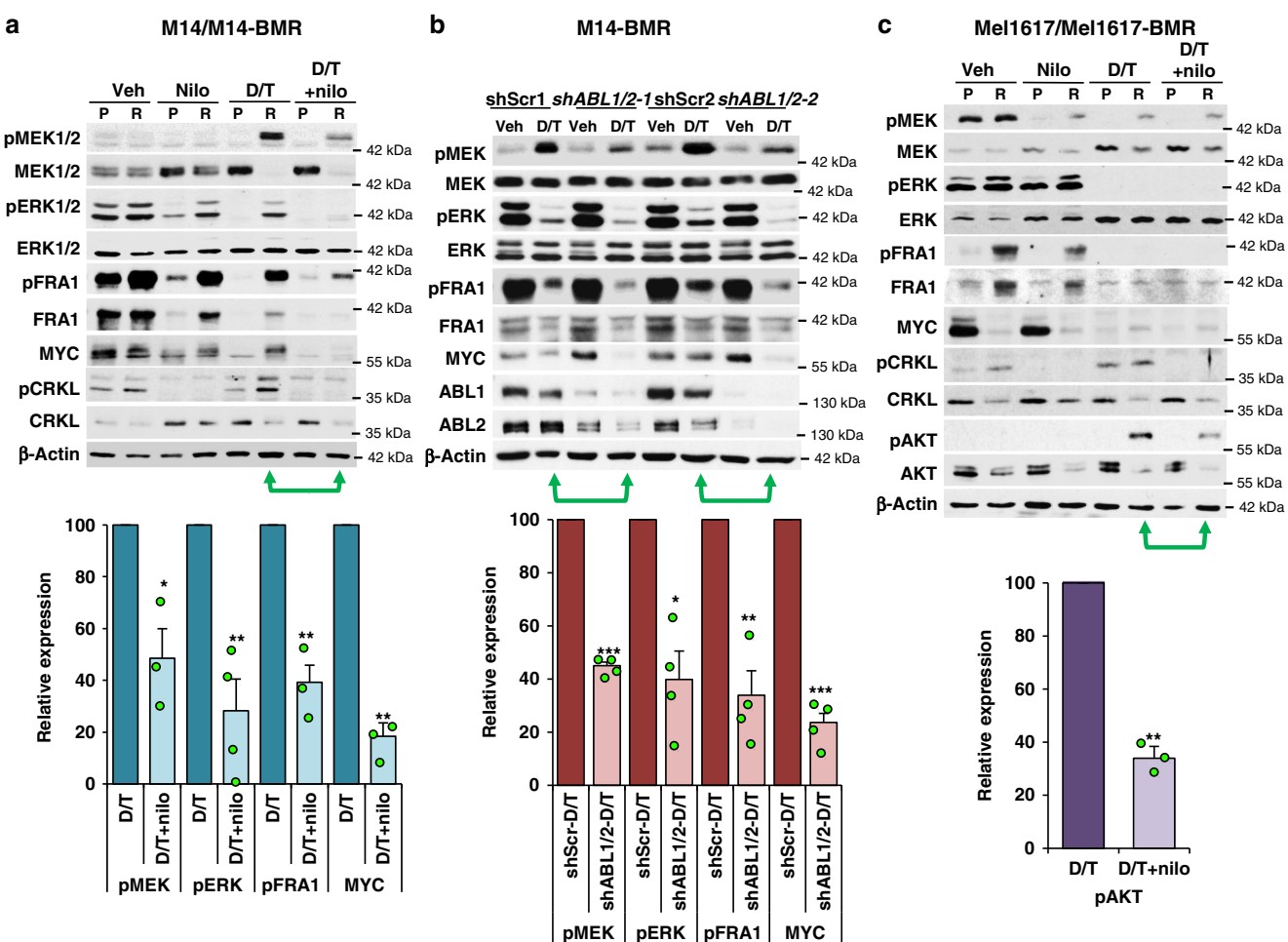

**Fig. 5 Nilotinib prevents reactivation of ERK-dependent (or ERK-independent) signaling during BRAFi/MEKi resistance. a, c** Parental or BRAFi/MEKi-resistant cells (-BMR) were treated with vehicle, nilotinib (nilo; 2.5 μM), dabrafenib/trametinib (D/T; BRAFi/MEKi; 50/10 nM) or the combination for 24 h, and the resulting lysates probed with the indicated antibodies. Quantitation of key signaling molecules for $n = 3$ independent experiments (except pERK, $n = 4$) is shown below; green arrows indicate lanes quantitated. Mean ± SEM *$p < 0.05$, **$p \leq 0.01$, ***$p < 0.001$ using two-sided, one-sample $t$-tests. Actual $p$ values (left→right): **a** 0.045, 0.0099, 0.0033, and 0.004. **c** **$p = 0.0045$. **b** M14-BMR cells expressing either vector (shScr) or IPTG-inducible shRNA targeting *ABL1* and *ABL2* (sh*ABL1/2*; two independent clones are shown) were treated with IPTG (1 mM) for 10 days prior to plating to induce expression, treated with DMSO (vehicle) or D/T (100 nM/20 nM) for 24 h, and lysates blotted with the indicated antibodies. $n = 4$ biological replicates (using two vector and two shRNA clones). **$p \leq 0.01$, ***$p < 0.001$ using two-sided, one-sample $t$-tests. Actual $p$ values (left→right): <0.0001, 0.011, 0.0056, and 0.000191. pCRKL (substrate of ABL1/2) is an indirect read-out of ABL1/2 activity and indicates the efficiency of nilotinib-mediated inhibition. For all experiments, BRAFi/MEKi-resistant lines were maintained in dabrafenib (100 nM) and trametinib (20 nM), but were cultured for 2 days without BRAFi/MEKi prior to plating for experiments.

reactivation (Fig. 6a). Since MEK/ERK reactivation during resistance induces MYC (a convergent point in multiple resistance pathways)[28], which is efficiently blocked by targeting ABL1/2, and *ABL1* mRNA is upregulated during resistance in primary melanomas, we examined the clinical significance of our findings by testing whether *ABL1* and *MYC* expression are linked in patient samples. Importantly, *ABL1* and *MYC* mRNAs are correlated in melanomas from patients prior to treatment (Fig. 6b, left) and in the TCGA dataset (Supplementary Fig. 5c), and an even more significant correlation was observed in paired samples from the same patients following relapse on BRAFi or BRAFi/MEKi (Fig. 6b, right). Thus, the resistance pathway we identified in cell lines also occurs in human patients.

**ABL1/2 activate MAP3Ks.** D/T treatment induces MEK1/2 (S217/S221) phosphorylation/activation, and silencing or inhibiting ABL1/2 reduces pMEK1/2 in the presence of PLX or D/T (Figs. 4a, b and 5a, b). Thus, ABL1/2 induce pMEK1/2 which

activates ERK1/2 and drives resistance. In BRAFi-resistant cells, RTKs can activate alternative MAP3Ks (CRAF, MAP3K8), which activate MEK1/2 thereby circumventing the effects of BRAFi[6,17,36]. CRAF directly phosphorylates/activates MEK, whereas MAP3K8 mRNA/protein upregulation activates MAP3K8 which directly activates MEK and ERK1/2[3,6,38]. MAP3K1, another MAP3K, primarily activates MAP2K4/MAP2K7 (MKK4/MKK7), MAP2Ks that activate JNK; however, it can phosphorylate MEK1/2 under some circumstances, and also can activate MEK/ERK indirectly via JNK feedback activation of ERBB RTKs[35,39]. To determine whether ABL1/2 regulate MAP3Ks, we examined whether MAP3Ks were activated/upregulated in resistant cells, and whether altering ABL1/2 activities prevents their upregulation. In BRAFi-resistant cells, CRAF (S338; PAK1 site) and MAP3K1 (T1402; activating residue in the catalytic domain) phosphorylation are induced and MAP3K8 expression is modestly increased, particularly in the presence of PLX (Supplementary Fig. 6a–d). Interestingly, the predominant phosphorylated (activated) form of MAP3K1 in melanoma cells is

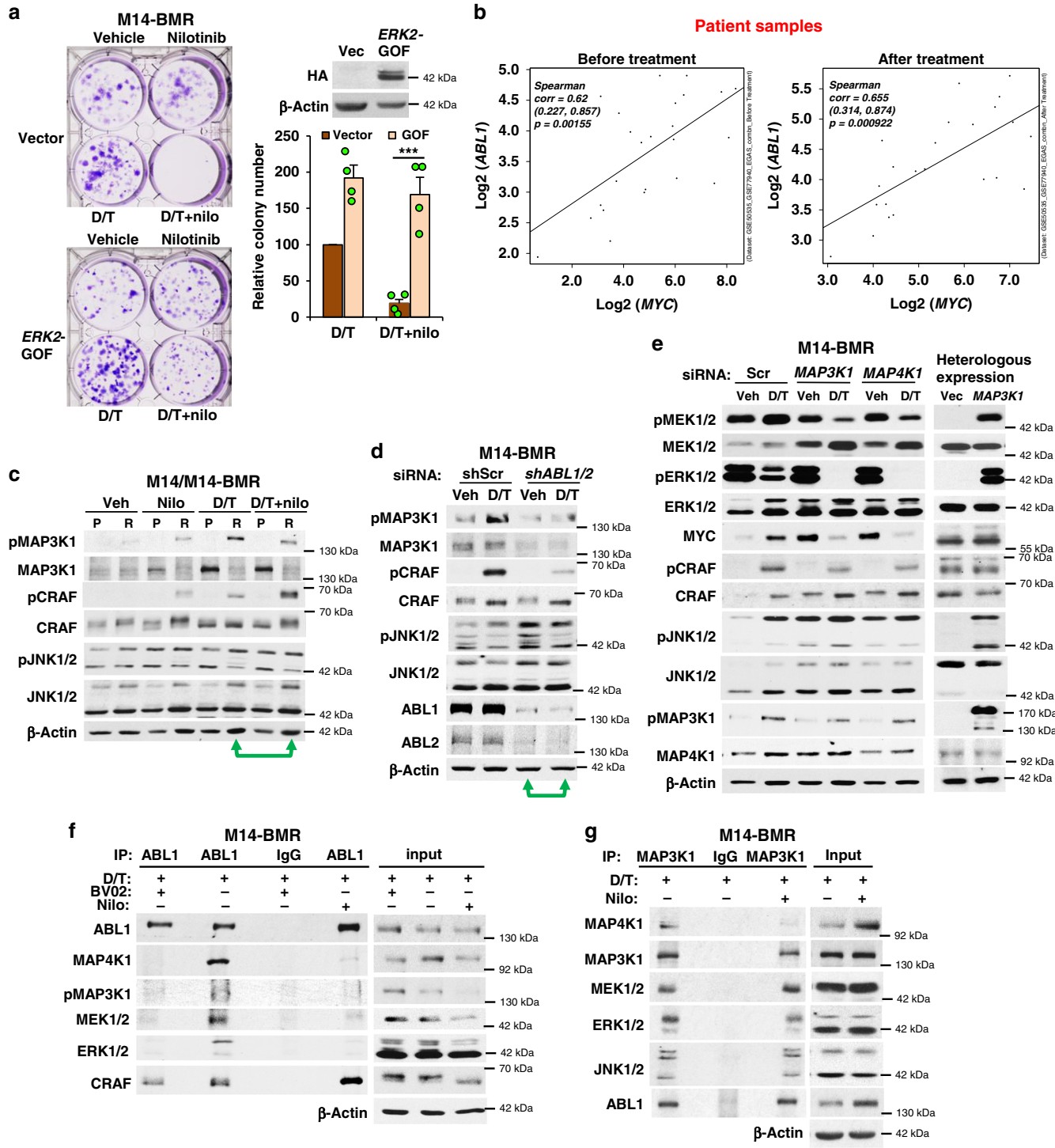

140 kDa, which is a known alternatively spliced form (NCBI website). Addition of nilotinib to PLX reduces pCRAF and pMAP3K1 in Mel1617-BR cells, whereas nilotinib alone reduces pMAP3K1 in 451-Lu-BR (Supplementary Fig. 6a–c). M14-BR cells harbor mutant *NRAS*, which directly binds/activates MAP3Ks[40]. Indeed, pCRAF and pMAP3K1 both are elevated in M14-BR cells; however, PLX + nilotinib treatment specifically reduces pMAP3K1 (Supplementary Fig. 6c, d). Consistent with nilotinib studies and with the notion that ABL1/2 regulates MAP3K1 in BRAFi-resistant cells, expression of constitutively active *ABL1/2* in BRAFi-sensitive cells increases MAP3K1 phosphorylation (Supplementary Fig. 6e).

In M14-BMR cells that have acquired BRAFi/MEKi resistance, MEK phosphorylation is induced in the presence of trametinib (MEKi), and this resistance mechanism has been postulated to be due to MEK phosphorylation by upstream kinases which causes reduced affinity of trametinib to the allosteric site[36]. MEK1 remains inhibited as reactivated ERK1/2 feeds back and phosphorylates/inhibits MEK1 (T292); however, this residue is not conserved in MEK2, and thus, MEK2 remains activated and drives resistance[36]. We found that M14-BMR cells harbor a mutation in *MAP4K1*, an upstream activator of MAP3K1 (Supplementary Data File 1)[39]. Consistent with this mutation being an activating event, MAP3K1 is upregulated

**Fig. 6 ABL1/2 drive ERK reactivation during resistance by promoting MAP3K activation. a** M14-BMR cells were engineered to express vector or HA-tagged constitutively active *ERK2* (GOF)[37]. *ERK*2-GOF expression is shown on the upper right. Vector or ERK2-GOF-expressing cells were plated, treated with D/T (50 nM/10 nM) in the absence or presence of nilotinib (nilo; 2.5 μM) for 7 days, wells washed, cells incubated in media lacking drugs for an additional 9 days, and stained with crystal violet. Quantitation of mean ± SEM for n = 4 independent experiments is shown on the right. ***p = 0.00084, two-sample t-test (two-sided). **b** Primary human paired melanoma datasets (RNAseq-77940, 50535, 65185, EGAS00001000992; microarray-50509) were analyzed for *ABL1* and *MYC* mRNA levels pre-/post-BRAFi, MEKi or BRAFi/MEKi treatment. Spearman's correlation coefficients were used to quantify correlations. Correlation (r), 95% confidence limits (in parentheses), and p values are shown. **c** M14-BMR or parental M14 cells were treated with the indicated drugs as in Fig. 5 for 24 h, and lysates blotted with antibodies. A representative experiment from n = 3 independent experiments is shown. Veh = vehicle. **d** M14-BMR cells expressing either vector (vec) or IPTG-inducible shRNA targeting *ABL1/2* were treated with IPTG (1 mM) for 5 days prior to plating, treated with DMSO (vehicle) or D/T (100 nM/20 nM) for 24 h, and lysates blotted with the indicated antibodies. A representative experiment from n = 4 independent experiments is shown. **e** M14-BMR cells were transfected with MAP3K1, MAP4K1, or scrambled siRNA, cells replated and treated with D/T (50/10 nM) −/+ nilotinib (2.5 μM) for 24 h, and lysates blotted with antibody (left). A representative experiment from n = 3 independent experiments is shown. (right) 293T cells were transfected with vector or MAP3K1 (exogenous form runs at 200 kDa), and lysates blotted after 48 h. A representative experiment from n = 2 independent experiments is shown. Vec = empty vector. **f, g** ABL1 (**f**) or MAP3K1 (**g**) immunoprecipitates (IP) from M14-BMR-treated cells, were blotted with the indicated antibodies. Representative experiments from n = 4 (**f**) and n = 3 (**g**) independent experiments are shown. Input is whole-cell lysate. Rabbit (**f**) or mouse (**g**) IgG served as a negative isotype controls. D/T = 100/20 nM; nilotinib = 2.5 μM; BV02 = 5 μM.

(phosphorylated) in M14-BMR cells, particularly in the presence of D/T (Fig. 6c). Silencing *ABL1/2* inhibits MAP3K1 activity (pMAK3K1-T1402) in the presence of D/T, and nilotinib/GNF-5 cooperate with D/T to reduce pMAP3K1 (Fig. 6c, d and Supplementary Fig. 6c, f, g). Although silencing *ABL1/2* reduces pCRAF, nilotinib paradoxically increases pCRAF in the presence of D/T, similar to our findings in M14-BR, but this does not result in increased pMEK/pERK or downstream signaling (Figs. 4a, 5a and 6c, d).

**MAP3K1 reactivates MEK/ERK during BRAFi/MEKi resistance**. ABL1/2 inhibition prevents MAP3K1 activation in all cell lines examined regardless of the mechanism of ERK reactivation, indicating that ABL kinases likely regulate MAP3K1. To identify the mechanism of regulation, we focused on the M14-BMR cell line, since these cells represent a clinically relevant model due to their resistance to both BRAFi and MEKi, a current treatment regimen. In breast cancer cells, MAP3K1 activates MEK1/2 indirectly by stimulating an MAP2K4/MAP2K7→JNK pathway, which results in JNK-mediated positive feedback activation of ERBB RTKs that activate MEK1/2[35]. However, in M14-BMR cells, silencing or inhibiting ABL1/2 (in the presence of D/T) inhibits MEK activity but does not reduce pJNK1/2; thus, ABL1/2 do not activate MEK via this mechanism (Fig. 6c, d). In other cell contexts, MAP3K1 phosphorylation of MEK1/2 does not activate ERK in cells[41]. However, we found that silencing MAP3K1 or MAP4K1 prevents MEK and ERK reactivation as well as downstream signaling (MYC) during resistance (+D/T) while having no effect on pJNK (Fig. 6e, left). Exogenous expression of MAP3K1 in a heterologous system (293T cells) activates MEK/ERK and JNK but has little effect on MYC (Fig. 6e, right). In contrast, in M14-BMR cells, exogenous MAP3K1 expression reduces nilotinib-mediated inhibition of pMEK and MYC, but has little effect on pJNK (Supplementary Fig. 6h). Thus, taken together our data demonstrate that ABL1/2 drive resistance by activating MAP3K1→MEK/ERK/MYC signaling, and although MAP3K1 has the potential to activate JNK or MEK, it preferentially activates MEK during BRAFi/MEKi resistance.

**ABL1-mediated coupling of MAP4K1/MAP3K1 to MEK/ERK requires interaction with 14-3-3 proteins**. Next, we investigated the mechanism by which ABL kinases impact MAP3K1 activity. Endogenous ABL1 coimmunoprecipitates (coIPs) with MAP4K1, pMAP3K1, CRAF, MEK, and ERK in M14-BMR cells exposed to D/T (Fig. 6f, second lane). We are not able to observe coIP of total MAP3K1 as all the antibodies we have tested are weak and

total MAP3K1 levels are low in resistant cells (Fig. 6c). The ABL2 antibody is inefficient at immunoprecipitating ABL2; however, we were able to observe endogenous coIP of ABL2 with MAP4K1, pMAP3K1, MEK, and CRAF (Supplementary Fig. 6i). Although nilotinib treatment (+D/T) increases the ability of the ABL1 antibody to IP ABL1, the binding of MAP4K1, MAP3K1, MEK, and ERK (but not CRAF) to ABL1 is efficiently reduced (Fig. 6f, fourth lane). Thus, ABL1/2, MAP4K1/pMAP3K1, and MEK/ERK are all in a complex, and ABL1/2 activity is required for the interaction.

ABL1 binding to 14-3-3 proteins sequesters ABL1 in the cytoplasm where it has a transforming function[20,21], and MAP3K1 also binds 14-3-3, although the consequence of this binding is unknown[42]. Thus, we tested whether 14-3-3 proteins mediate binding between ABL1, MAP4K1/MAP3K1, and MEK/ERK. Importantly, treatment with the 14-3-3 interaction inhibitor, BV02, prevents binding of MAP4K1, pMAP3K1, MEK1, and ERK to ABL1 (Fig. 6f, first lane). Thus, interaction with 14-3-3 proteins is critical for formation of the ABL1-containing complex. Indeed, ABL1, MAP4K1, pMAP3K1, MEK, ERK, and CRAF all coIP with 14-3-3 proteins; however, BV02 treatment only prevents ABL1, pMAP3K1, and to a lesser extent, MAP4K1 from binding 14-3-3 (Supplementary Fig. 6j), indicating that these proteins mediate the interaction. Similar to nilotinib, silencing ABL1/2 reduces binding of MAP4K1, MAP3K1, MEK, and CRAF to 14-3-3, but increases the binding of JNK, indicating that ABL1/2 are required for formation of MAP4K1/MAP3K1/MEK but not MAP4K1/MAP3K1/JNK complexes (Supplementary Fig. 6k). Significantly, silencing or inhibiting (nilotinib) ABL1/2 uncouples binding of MAP4K1 to MAP3K1 during ERK reactivation-driven resistance (Fig. 6g and Supplementary Fig. 6l). Numerous MAP4Ks have the ability to phosphorylate MAP3K1, and MAP4Ks can phosphorylate a plethora of MAP3Ks[39]. Thus, these data are significant as they indicate that ABL1/2 not only drive signaling of MAP3K1 to MEK/ERK (rather than to JNK/JUN) but also mediate coupling of the upstream kinase, MAP4K1, specifically to MAP3K1.

**ABL1/2 phosphorylate MAP3K1 and 14-3-3-ε**. Since ABL1/2 activity is required for assembly of the ABL1/MAP4K1/MAP3K1/MEK/ERK complex and for coupling of MAP4K1 to MAP3K1, ABL1/2 likely phosphorylate one or more members of the complex. Thus, we examined whether ABL1/2 phosphorylate MAP3K1 and/or 14-3-3 proteins, since MAP3K1 has been implicated as a target of ABL1 in other systems[43–45], and tyrosine phosphorylation of 14-3-3 is known to regulate its binding to other proteins[46]. Co-expression of MAP3K1 and activated ABL1

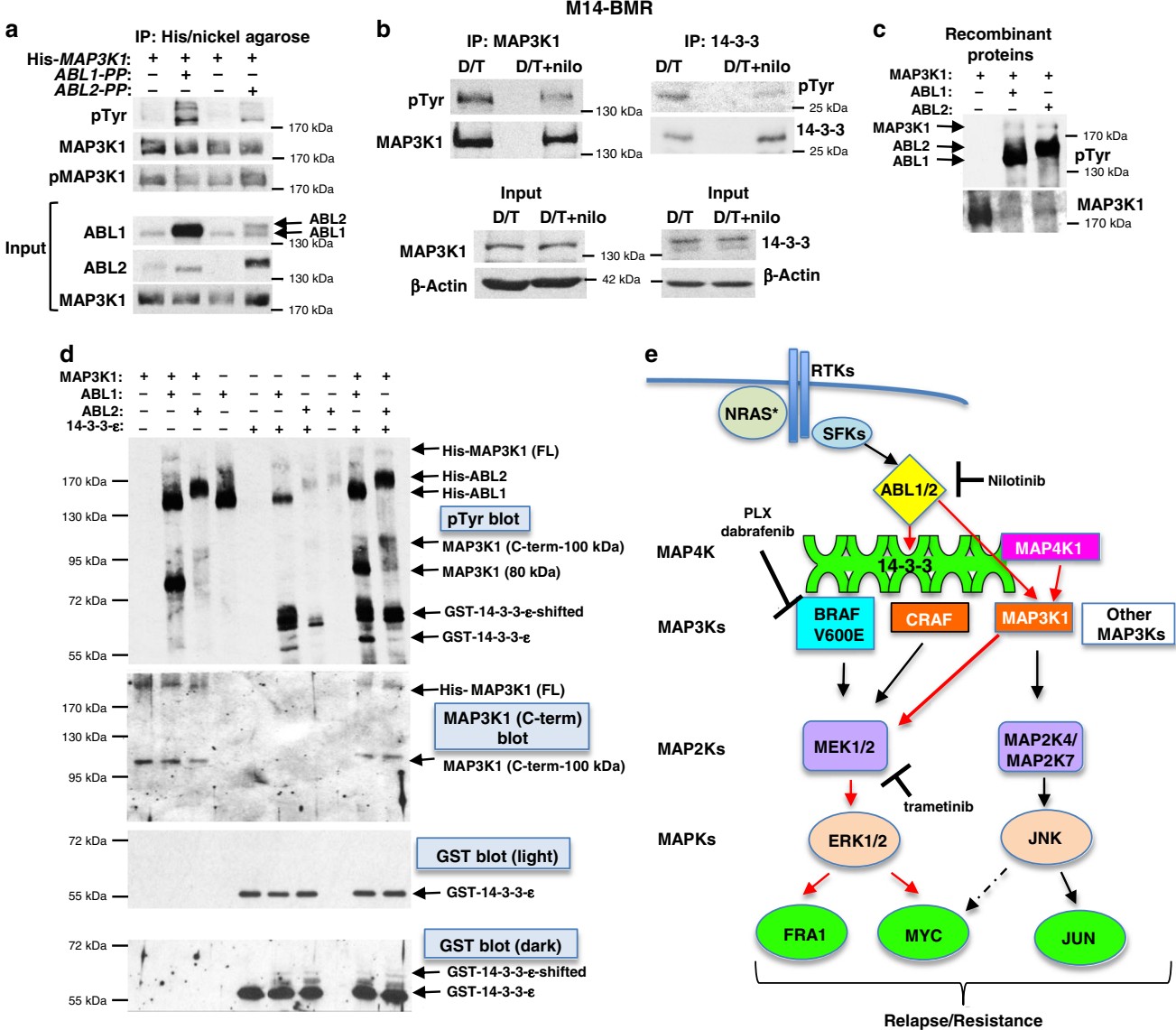

**Fig. 7 ABL1/2 phosphorylate MAP3K1 and 14-3-3-ε. a** 293T cells were transfected with His-*MAP3K1* and/or constitutively active forms of *ABL1* or *ABL2* (PP), His-*MAP3K1* pelleted with nickel agarose, complexes blotted with phospho-tyrosine (pTyr) antibody, and blot stripped and reprobed with control antibodies. Blots shown are representative of n = 3 independent experiments. **b** Endogenous MAP3K1 or 14-3-3 (pan) were immunoprecipitated from M14-BMR cells, and IPs blotted with the indicated antibodies. Blots shown are representative of n = 2 independent experiments. **c** Recombinant forms of ABL1 or ABL2 were incubated with full-length, recombinant MAP3K1 in a "cold" in vitro kinase assay, and reactions blotted with phospho-tyrosine (pTyr) antibody, stripped and reprobed with antibody to MAP3K1. Blots shown are representative of n = 2 independent experiments. **d** Recombinant proteins were incubated in a kinase assay as in **c** using either His-MAP3K1, GST-14-3-3-ε, or the combination as substrates. The phospho-tyrosine blot (top) was stripped and reprobed with antibody directed at the MAP3K1 C-terminus (middle) or GST (bottom; to visualize 14-3-3). Blots shown are representative of n = 2 independent experiments. **e** Scheme showing mechanism by which ABL1/2 drive ERK reactivation and resistance (red arrows).

or ABL2 in a heterologous system results in tyrosine phosphorylation of MAP3K1 (Fig. 7a). Moreover, nilotinib treatment of M14-BMR cells (in the presence of D/T) reduces MAP3K1 and 14-3-3 tyrosine phosphorylation (Fig. 7b). To confirm that ABL1/2 phosphorylate MAP3K1 and 14-3-3, and to assess whether the phosphorylation is direct, we performed kinase assays using recombinant, purified proteins. Incubation of MAP3K1 with ABL1 or ABL2 results in modest tyrosine phosphorylation of full-length MAP3K1 (Fig. 7c). The supplied recombinant MAP3K1 has some degradation products (see Origene website). Interestingly, ABL1 efficiently phosphorylates the 80 kDa breakdown product, whereas ABL1 and ABL2 phosphorylate a 100 kDa breakdown product recognized by a C-terminal MAP3K1 antibody (Fig. 7d, left and Supplementary Fig. 7a, left). To test

whether ABL1/2 also phosphorylate 14-3-3, and whether the presence of 14-3-3 proteins acts as a scaffold to enhance ABL1/2-mediated phosphorylation of MAP3K1, we incubated ABL1/2 with GST-14-3-3-ε, a 14-3-3 family member known to bind ABL1 and MAP3K1[20,21,42]. Importantly, ABL1 efficiently phosphorylates 14-3-3-ε, and the presence of MAP3K1 enhances the ability of ABL2 to phosphorylate 14-3-3-ε and vice-versa (Fig. 7d, right side). Interestingly, ABL1/2 phosphorylate numerous sites on 14-3-3-ε as increased incubation time (30′ versus 15′) induces a significant band shift which is observed both in phospho-tyrosine and GST blots (Fig. 7d and Supplementary Fig. 7a, b). Thus, ABL1/2 phosphorylate MAP3K1 and 14-3-3, and ABL1/2 activity is required to assemble the ABL1/MAP4K1/MAP3K1/MEK/ERK signaling complex, which results in MAP4K1 activation of

MAP3K1, and subsequent MEK/ERK reactivation during D/T resistance (Fig. 7e).

**Nilotinib reverses resistance, in vivo.** To determine whether nilotinib could potentially be utilized as an agent to reverse or prevent resistance in the clinic, we performed three in vivo studies. First, Mel1617 parental or BRAFi-resistant xenografts (-BR) were established (200 mm$^3$) in SCID-beige mice, and mice bearing parental tumors were treated with vehicle or PLX4720 (to assess BRAFi-mediated regression of sensitive tumors), whereas mice bearing the resistant variant were treated with vehicle, nilotinib, PLX4720 or PLX + nilotinib. Tumor regression was efficient in parental xenografts treated with PLX (Fig. 8a, open circles), whereas resistant tumors in mice treated with vehicle, nilotinib or PLX grew quickly and reached the endpoint in 11 days (Fig. 8a). In contrast, overall survival was much higher in mice treated with the PLX + nilotinib combination (Fig. 8b). Interestingly, mice treated with PLX + nilotinib segregated into two groups (responders $n = 7$, non-responders $n = 4$; Fig. 8a, c). Resistant tumors completely regressed in responding mice, similar to PLX-treated parental tumors, whereas tumor growth curves of non-responding mice were no different from controls (Fig. 8a, c). Thus, PLX + nilotinib, which did not affect mouse weights (Supplementary Fig. 8a), was effective in the majority of mice.

In order to gain mechanistic insight into why some mice responded well to the combination regimen while others did not, responding and non-responding tumors (two apiece) were subjected to WES. There were 84 gene mutations in common between the four tumors (including MAP2K2-Q60P): 26 genes unique to both responding tumors and 16 mutations unique to both non-responding tumors (Supplementary Data Files 3–8). Of note, both non-responding tumors had a mutation within the Rho-GEF domain of VAV3, a Rac-GEF and downstream substrate of BCR-ABL (Supplementary Data Files 3, 6 and 7)[47]. Moreover, both responding tumors had the same mutation in ABI2 (T292S), an ABL1 interacting protein and regulator of ABL1 activity/signaling[48]. Furthermore, the protein phosphatase N4 (PTPN4), which binds ABL1 and regulates CRK-I (ABL substrate) phosphorylation[49], was mutated in responding and non-responding tumors; however, the mutation observed in the responding tumors occurred in a different functional domain from the mutation observed in the non-responding tumors, and thus, potentially, could have opposite effects (Supplementary Data Files 3, 8 and 9). Thus, responding and non-responding tumors likely originated from two independent cell populations (even though Mel1617-BR was described as clonal)[17], and non-responding tumors likely did not respond to nilotinib+PLX treatment due to mutations that resulted in activation of ABL1/2 downstream substrates.

To determine whether nilotinib also reverses resistance to BRAFi/MEKi, in vivo, parental M14 or BRAFi/MEKi-resistant (M14-BMR) xenografts were established in nude mice. Mice harboring parental tumors were treated with vehicle or dabrafenib+trametinib (D/T), whereas mice harboring M14-BMR tumors were treated with vehicle, nilotinib, D/T, or D/T + nilotinib. Parental tumors grew very quickly and D/T treatment induced rapid regression (Fig. 8d, orange and red lines). For M14-BMR xenografts, D/T only modestly slowed tumor growth, and nilotinib alone had no effect, whereas the triple-combination (D/T + nilotinib) induced efficient tumor regression (Fig. 8d). Four mice from D/T and D/T + nilotinib groups and all vehicle- and nilotinib-treated mice were euthanized on day 35 due to large tumor sizes in vehicle/nilotinib groups. Triple-combination-treated tumors were small and consisted of mostly matrigel with

occasional small nests of cells (Fig. 8d, e and Supplementary Fig. 8b), and Kaplan–Meier curves demonstrated elevated overall survival (Fig. 8f). D/T + nilotinib-treated tumors also had reduced expression of MYC, consistent with our data in cell lines and patient samples indicating that ABL kinases drive MYC expression during resistance (Fig. 8e and Supplementary Fig. 8c, d). The remainder of the D/T and D/T + nilotinib mice ($n = 8$/group) were followed long term. Prolonged tumor regression was observed in the triple-combination-treated group, whereas all D/T-treated animals had to be euthanized due to large tumor size ($\cong$1000–1500 mm$^3$) by day 81 (Fig. 8g and Supplementary Fig. 8e). Moreover, D/T-treated tumors were highly aggressive as there was a high rate of lymph node (LN) metastases (75%), whereas addition of nilotinib to the D/T regimen completely blocked metastatic progression (Supplementary Table 1).

We continued to follow the triple-combination mice, and found that 25% (2/8) never exhibited relapse even after following for 160 days (Supplementary Fig. 8e). The rest of the mice relapsed between days 95–100 (Supplementary Fig. 8e). CML patients develop resistance to nilotinib or imatinib by acquiring kinase domain mutations in ABL1 (e.g. gatekeeper mutation, T315I), which renders the protein insensitive to the drugs[50]. To test whether the resistance we observed was likely due to ABL1 or ABL2 mutation, we replaced nilotinib with ponatinib, a third-generation ABL inhibitor that inhibits the gatekeeper mutation[50]. Unfortunately, due to toxicity of the ponatinib+D/T combination, we were unable to treat mice daily; we often had to withhold treatment due to anorexia; and the treatment had to be discontinued after 10 days. However, despite these difficulties, ponatinib induced tumor regression in all treated mice (Supplementary Fig. 8f), indicating that the mechanism of D/T + nilotinib resistance may involve ABL1 or ABL2 mutation. Thus, nilotinib reverses resistance to dabrafenib/trametinib, a common event in the clinic, substantially extending survival. Importantly, no obvious toxicity was noted for the D/T + nilotinib regimen, and there was no significant change in animal weights despite treatment for >100 days (Supplementary Fig. 8g).

**Nilotinib prevents resistance from developing, in vivo.** The prior data indicate that nilotinib may be an effective add-on treatment for patients who develop resistance to BRAFi/MEKi. To test whether nilotinib also prevents resistance from developing in the first place, and thus, whether it could potentially be used in BRAFi/MEKi-naïve patients, we utilized an in vivo resistance model. M14 parental cells that are sensitive to BRAFi/MEKi were established as xenografts in nude mice. Once tumors were well-established (mean volume = 200 mm$^3$), mice were treated with vehicle, nilotinib, D/T, or D/T + nilotinib. Nilotinib alone had no effect on tumor growth (Fig. 8h), whereas D/T induced rapid regression followed by relapse and resistance around day 40 (Fig. 8h and Supplementary Fig. 9a). In contrast, D/T + nilotinib-treated animals had prolonged tumor regression with only 8% (1/12) of animals relapsing (defined as having tumors ≥300 mm$^3$) within the first 70 days of treatment and only 17% (2/12) relapsing by day 100 as compared to 100% relapse for D/T-treated mice by day 69 (Fig. 8h, i and Supplementary Fig. 9a). Moreover, the triple-combination mice had increased progression-free survival (Supplementary Fig. 9b). Furthermore, as in the resensitization experiment, the triple combination had no noticable toxicity, and mice weights were not significantly different between D/T and D/T + nilotinib groups even after 100 days treatment (Supplementary Fig. 9c). Thus, nilotinib not only resensitizes melanomas to BRAFi/MEKi, but also prevents resistance from developing in the first place.

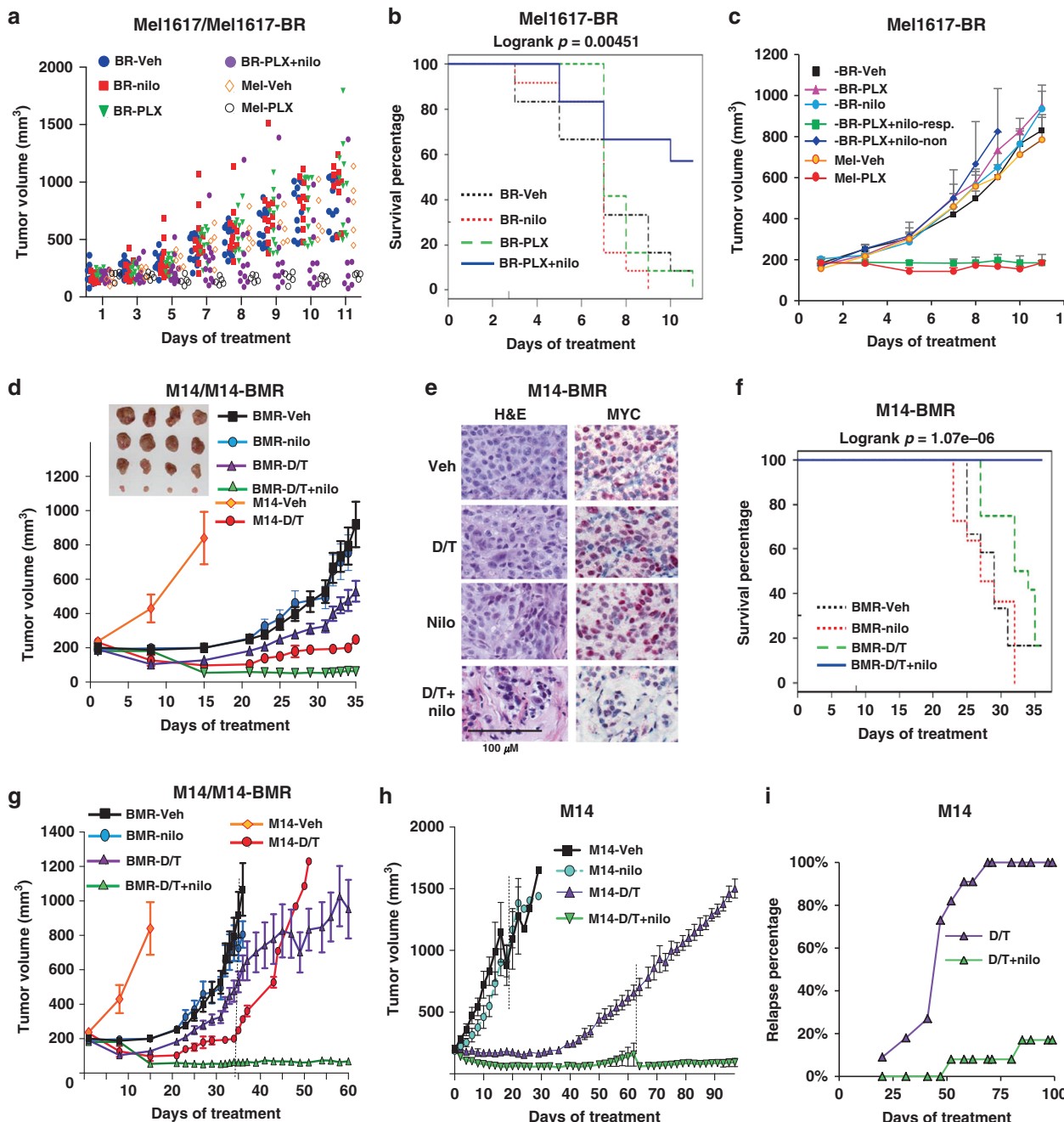

**Fig. 8 Nilotinib reverses and prevents resistance, in vivo. a–c** Mel1617-BR (BR) or parental cells (Mel) were injected s.q. into SCID-beige mice, and when average tumor size was 200 mm³, Mel1617-BR mice were randomized to vehicle (Veh, n = 12), nilotinib (nilo, n = 12; 33 mg/kg, b.i.d.), PLX4720 (PLX, 25 mg/ kg, n = 12), or combination (n = 11). Parental xenografts were treated with vehicle or PLX (n = 5/group). Mice were euthanized when control tumors ≅1500 mm³ (or were ulcerated). **a** Individual mouse tumor volumes over time. **b** Kaplan–Meier survival analysis comparing tumor doubling across groups. Percentage of mice not reaching the tumor doubling endpoint. Logrank p = 0.00451. PLX + nilo vs. vehicle, p = 0.0202; PLX + nilo vs. nilo, p = 0.0093; PLX + nilo vs. PLX, p = 0.0202 using pairwise comparisons and Holm's multiple comparison adjustment. **c** PLX + nilotinib-treated mice were divided into responding (n = 7) and non-responding (n = 4) groups, and mean ± SEM tumor volumes plotted. Some SEMs are too small to visualize. One-way ANOVA followed by Sidak's multiple comparisons test for responding vs. other groups. veh, p = 0.0104; nilo, p = 0.0004; PLX, p = 0.0027; non-responding, p = 0.0116. **d–g** M14-BMR (BMR) or parental M14 (M14) xenografts were established in nude mice. BMR xenografts treated with vehicle, nilotinib (33 mg/kg, b.i.d.), dabrafenib (25 mg/kg/day) + trametinib (0.15 mg/kg/day) (D/T), or D/T + nilotinib (n = 12/group) when tumors were 200 mm³. Parental xenografts were treated with vehicle or D/T (n = 5/group). **d** Mean ± SEM tumor volumes. Vehicle/nilotinib-treated mice and four mice from D/T and D/T + nilo groups were euthanized on d35. Linear mixed model analysis: for all comparisons, p = 3e−04. **e** H&E and MYC immunohistochemical staining of representative tumors (**d**) from n = 4 mice/group. **f** Survival analysis. Time-to-tumor doubling on d35. Percentage of mice not reaching the tumor doubling endpoint. Logrank p = 1.07e−06. Pairwise comparisons and Holm's: p < 6.46e−05 for all groups. **g** D/T and D/T + nilotinib mice (n = 8/group) were followed long term. Mean ± SEM tumor volumes are shown. Dotted line indicates loss of euthanized mice (4/group). **h, i** In vivo prevention study. M14 parental tumors were established in nude mice, and mice treated with the indicated drugs. Vehicle, n = 5; nilo, n = 6; D/T, n = 11; DT + nilo, n = 12. **h** Mean ± SEM tumor volumes. Dotted lines indicate loss of mice (high tumor volume and/or tumor ulceration). **i** Percentage of mice that developed resistance/relapse (tumors ≥ 300 mm³). Fisher's exact test, p = 0.0000673 (two sided). Additional data and statistics related to these experiments are shown in Supplementary Fig. 8.

## Discussion

Despite the development of immunotherapies and BRAFi/MEKi, the 5-year survival rate for patients with metastatic melanoma remains low (23%; Cancer.Net). Thus, there is an unmet need for new drug combinations to extend survival. In this work, using loss-of-function, gain-of-function and rescue approaches, as well as cell lines, patient samples, and in vivo approaches, we demonstrate that ABL1/2 drive resistance to BRAFi and BRAFi/MEKi by activating MEK/ERK/MYC via alternative MAP3K pathways. Moreover, we show that ABL1/2 play a critical role in linking MAP4K1/MAK3K1 to MEK/ERK reactivation during BRAFi/MEKi resistance. Furthermore, targeting ABL1/2 using an FDA-approved ABL1/2 inhibitor, nilotinib, resensitizes resistant cells to BRAFi or BRAFi/MEKi, inhibiting survival and promoting apoptosis, in vitro, inducing prolonged regression of BRAFi/MEKi xenografts, in vivo, and preventing resistance from developing in the first place. Thus, these studies indicate that repurposing nilotinib, an FDA-approved drug used to treat leukemia for more than a decade, has the potential to prolong melanoma patient survival in both MAPKi-refractory and treatment-naïve settings.

We provide numerous lines of evidence to demonstrate that ABL1/2 drive resistance. (1) ABL1/2 are activated in resistant lines. (2) ABL1 mRNA is elevated in melanomas from patients that relapse on BRAFi or BRAFi/MEKi therapy. (3) ABL1 cytoplasmic/nuclear ratio and ABL1-T735 phosphorylation (14-3-3 binding site) are increased in resistant cells, consistent with findings demonstrating that cytoplasmic retention drives ABL1 oncogenic function[9]. (4) Constitutively active forms of ABL1/2 drive resistance in treatment-naïve melanoma cells, and induce MEK/ERK reactivation. (5) Silencing ABL1/2 (and subsequent rescue with ABL1/2 addback), use of an allosteric inhibitor that targets only ABL1/2 (GNF-5), and two different ABL1/2 TKIs (nilotinib, ponatinib) reverse BRAFi/MEKi resistance and prevent MEK/ERK reactivation. (6) Expression of nilotinib-resistant, mutant forms of ABL1/2 rescues nilotinib's anti-survival effects. Importantly, although the genomic mechanisms of resistance are different in each line, nilotinib reverses resistance in all lines queried, indicating that ABL kinases are key signaling nodes during resistance. Additionally, MYC upregulation is a convergent point for multiple resistance pathways[28], and we show that ABL1/2 drive MYC upregulation during resistance using cell lines, animal models, and patient samples. Notably, our data also are consistent with findings identifying ABL1 in a kinome-wide screen for overexpressed kinases that can predict resistance to BRAFi/MEKi[51]. However, despite these compelling data, due to the fact that ABL1/2-T315I introduction into M14-BR cells also increases viability in the absence of nilotinib (a limitation to this assay; Supplementary Fig. 2g), we cannot completely rule out the possibility that some of nilotinib's effects are mediated by non-ABL1/2 nilotinib targets.

ERK pathway reactivation is a common mechanism of MAPKi resistance[3,5], and most cell lines we queried acquired resistance in this manner; however, the mechanisms by which ERK is reactivated are distinct. We identified activating mutations in MAP2K2/MEK2 (Mel1617-BR) and NRAS (M14-BR), which are resistance mechanisms documented in patient samples[5], indicating our lines are clinically relevant. The mechanisms of ERK reactivation in M14-BMR and 451-Lu-BR were more elusive. In M14-BMR, BRAFi/MEKi activate MEK1/2 indicating that upstream kinases phosphorylate MEK1/2, which reduces trametinib binding, leading to subsequent MEK2-driven resistance[36]. NextGen sequencing revealed that M14-BMR harbor a MAP4K1 (P422Q) mutation that has also been identified in a melanoma patient (cBioportal website), and the residue is adjacent to a critical serine phosphorylation site (S421; PhosphoSite). Phosphorylation of MAP3K1, a downstream substrate of MAP4K1, is

increased in these cells consistent with the MAP4K1 mutation being gain-of-function. Interestingly, we show that MAP3K1 is activated in all resistant lines driven by ERK reactivation (independent of the ERK activation mutation), particularly in the presence of BRAFi or BRAFi/MEKi, and nilotinib alone (451-Lu-BR) or nilotinib in combination with MAPK inhibitors (other 3 lines) potently inhibits MAP3K1 phosphorylation/activation. Our mechanistic data provide insight into how/why this occurs by demonstrating that MAP4K1/MAP3K1 induce MEK/ERK rather than JNK/JUN activation during resistance. Moreover, ABL1/2 activity and 14-3-3 proteins are critical for the assembly of MAP4K1/MAP3K1/MEK/ERK complexes, and ABL1/2 drive coupling of MAP4K1 to MAP3K1. Our data are consistent with data obtained in other cell contexts showing that ABL1 interacts with and/or phosphorylates MAP4K1 and MAP3K1[43,45,52–54], and with the identification of increased MAP3K phosphopeptides in imatinib-resistant BCR-ABL1-expressing CML cells (indicates they are likely substrates)[55].

The IGF1 receptor is activated in 451-Lu-BR and Mel1617-BR cells[17]; however, nilotinib alone is highly effective at reducing viability in 451-Lu-BR, whereas nilotinib cooperates with BRAFi in Mel1617-BR. Interestingly, pCRAF is elevated in Mel1617-BR, but not in 451-Lu-BR. Moreover, Mel1617-BR has a MAP2K2/MEK2 mutation (Q60P; analogous to Q56P in MEK1). MEK1-Q56P is weakly transforming and must coexist with other lesions that activate CRAF (e.g. IGF1-R) in order to fully activate ERK and have complete transforming activity[56]. In contrast, 451-Lu-BR harbors a mutation in the JNK-binding domain of MAP2K7 (substrate of MAP3K1; cooperates with MAP2K4 to activate JNK). Breast cancer cells harboring loss-of-function mutations in MAP2K4 (and MAP3K1) lack JNK-mediated activation of ErbB→CRAF→MEK1/2, and thus, are highly sensitive to MEKi[35]. Loss-of-function mutations in MAP2K7 would be predicted to act similarly since MAP2K7 and MAP2K4 cooperate to activate JNK. Indeed, 451-Lu-BR cells are highly sensitive to trametinib, in contrast to Mel1617-BR cells, which display collateral resistance to trametinib (Supplementary Fig. 10), and 451-Lu-BR are highly sensitive to nilotinib, which also reduces pMEK1/2. Furthermore, pJNK is repressed in 451-Lu-BR cells (compared to the parental line), which is consistent with 451-Lu-BR cells harboring a MAP2K7 loss-of-function mutation (Supplementary Fig. 6b). Thus, loss-of-function mutations in MAP2K7 (and also likely MAP2K4 and MAP3K1) could potentially be biomarkers for resistant tumors that are highly sensitive to nilotinib alone, and thus may identify patients that could benefit from monotherapy; future experiments will address this exciting possibility.

The data shown here have high clinical and translational relevance. In the in vivo resensitization assay, D/T + nilotinib significantly increased survival, and in the in vivo prevention study, 83% of mice never developed resistance to the triple combination even after 100 days of treatment. Nilotinib has been FDA-approved since 2007; its safety/toxicity profile is known; a 400 mg (b.i.d.) dose, which is used for drug-resistant CML, results in 1.95 μM plasma concentrations, well within the range of our in vitro doses; and our in vivo dose (33 mg/kg) is the same or lower than doses utilized by others in CML models[57–59]. Moreover, mice treated with the triple combination exhibited no noticeable toxicity despite use for 100–160 days. Taken together, these data indicate that the triple combination may be an effective therapy for treatment-refractory BRAF-driven melanomas. Indeed, we are currently planning a Phase I trial to test the safety and efficacy of the triple combination in the second-line setting. If positive, this may lead to critical prevention studies, where the combination has the potential to increase survival of melanoma patients from the outset.

## Methods

**Reagents.** *Cell lines*: M14 cells were obtained from NCI (NCI-60 panel) in 2016. Mel1617, Mel1617-BR, 451-Lu, 451-Lu-BR (c5), and WM164 were obtained from Meenhard Herlyn (Wistar Institute, Philadelphia, PA) via an MTA in 2011. Mel1617-BR and 451-Lu-BR were described as clonal lines, established by culturing cells in increasing doses of the BRAFi, SB590885, followed by isolation of individual clones[17]. Resistant lines were maintained in SB590885 (1 μM). Short tandem repeat (STR) analysis was repeated on the four cell lines in 2018 (Wistar Institute, Philadelphia, PA). Polyclonal M14-BR, M14-BMR, and Mel1617-BMR lines were generated in our lab from M14 and Mel1617 parental cells by culturing cells in increasing concentrations of the vemurafenib analog, PLX4720 (M14-BR; up to 1 μM) or dabrafenib+trametinib (M14-BMR; Mel1617-BMR up to 100 nM/20 nM) until resistant at the indicated dosage (≅6 months), and pooling resistant clones. M14-BR was maintained in PLX (1 μM), whereas -BMR lines were maintained in D/T (100 nM/20 nM).

*Antibodies*: See Supplementary Table 2 for commercial antibodies and their use. The ABL2 antibody used for coIP and kinase assays was previously described[18,19].

*DNA constructs and stable expressing cell lines*: pBabe-puro-HA-ERK2-GOF (D231N, R67S; plasmid #53203) deposited by Chris Counter (Duke University, Durham, NC), GST-14-3-3-ε (#13279) deposited by Michael Yaffe (MIT, Boston, MA), and His-MAP3K1 (#12181) deposited by Gary Johnson (UNC, Chapel Hill, NC) were obtained from Addgene (Watertown, MA). M14-BMR cells expressing HA-ERK2-GOF were obtained by transfecting either the vector or *ERK2*-GOF into M14-BMR cells using Lipofectamine 2000 (Invitrogen; manufacturer's instructions), selecting clones with puromycin (0.5 μg/ml), picking individual clones, and screening for expression by blotting with HA antibody. Positive clones were pooled. Constitutively active *ABL1* and *ABL2* (PP)[26] were removed from their vectors (pSGT-*ABL1*-PP, BamH1[26]; PK1-*ABL2*-PP, *Eco*R1/*Xho*I[60]) blunted and cloned into the *Swa*I site of PiggyBac cumate-inducible transposable vector (Systems Bioscience; Palo Alto, CA). Vector or *ABL1/2* constructs were stably transfected into WM164 cells using Lipofectamine 2000, selected with puromycin (1.2 μg/ml), and clones were pooled (polyclonal population). FACS analysis indicated >90% of cells were GFP positive. Expression was observed in the absence (leaky) and presence of cumate. Low-level expression can be observed via phospho-CRKL blotting, which is an indirect measurement of ABL1/2 activity. M14-BR cells expressing nilotinib-resistant ABL1/2 mutant (T315I) forms[61] were established by transfecting ABL1/2-T315I, cloned into the Piggybac Transposon vector (see above), followed by puromycin selection (1.2 μg/ml) and pooling resistant clones.

*siRNA and shRNA-expressing cell lines*: MAP3K1 (s8669) and MAP4K1 (s22080) silencer select siRNAs (or scrambled control #1) were obtained from ThermoFisher (Waltham, MA). Cells expressing IPTG-inducible shRNA targeting *ABL1* and *ABL2* (PLK01-IPTG-3XLacO vector; custom synthesis-Millipore Sigma, St. Louis, MO; GGGAAATTGCTACCTATGG) were obtained following lentiviral infection and selection with puromycin (0.5 μg/ml). Clones were picked, expanded, and screened for knockdown by western blot. Inducible shRNA-expressing cells were treated with IPTG (1 mM; Sigma; 6 days) prior to screening. Cells were treated with IPTG (1 mM) for 5–10 days prior to plating for CellTiter Glo, colony-forming assays, or western blots.

*Drugs*: GNF-5 and SB590885 were purchased from Selleck (Houston, TX), whereas ponatinib, nilotinib (for in vitro experiments and for PLX4720 + nilotinib in vivo experiments), and trametinib were purchased from MedChem Express (MCE, Monmouth Junction, NJ). Nilotinib (for D/T + nilotinib in vivo experiments) and dabrafenib were kindly provided by Novartis (Switzerland) under a Materials Transfer Agreement (MTA). PLX4720 for in vitro and in vivo use was obtained under an MTA with Plexxikon Inc. (Berkely, CA). SU6656 was purchased from Calbiochem (VWR; Atlanta, GA). BV02 was from MilliporeSigma.

### Purification of GST-fusion proteins (GST-CRK, GST-14-3-3-ε).
Log-phase BL21 bacteria transformed with the constructs were induced with IPTG (0.1 mM) for 5 h, pellets were lysed in MT-PBS (150 mM NaCl, 16 mM $Na_2HPO_4$, 4 mM $NaH_2PO_4$, 50 mM EDTA, pH 7.3 + inhibitors—see below), Triton X was added (1% final), supernatant was incubated with glutathione sepharose for 30′, beads were washed 3× in MT-PBS followed by elution buffer lacking glutathione (100 mM Tris, pH 8, 120 mM NaCl), and eluted in elution buffer containing 15 mM reduced glutathione. Elutions were pooled, dialyzed against phosphate-buffered saline (Slide-A-Lyzer), and quantitated via SDS/PAGE comparing to a known standard (BSA).

### In vitro kinase assay.
*ABL1/2 assay*: Subconfluent cells were serum-starved for 16 h, and lysed in kinase lysis buffer [50 mM HEPES pH 7.0, 150 mM NaCl, 10% glycerol, 1% Triton X-100, 1.5 mM $MgCl_2$, 1 mM EGTA, and fresh protease and phosphatase inhibitors (leupeptin 10 μg/ml, pepstatin 10 μg/ml, aprotinin 10 μg/ml, PMSF 1 mM, NaF 25 mM, sodium orthovanadate 1 mM)]. Lysates were precleared with rabbit IgG (ABL1) or normal rabbit serum (ABL2) and protein-A sepharose. ABL1 or ABL2 was immunoprecipitated from cellular lysates (30–50 μg ABL1; 60–100 μg ABL2) using agarose-conjugated K12 antibody for ABL1 or an ABL2-specific antibody[19] and protein-A sepharose for ABL2. IPs were washed twice in RIPA buffer, twice in NaCl buffer (10 mM Tris, pH 7.4, 5 mM EDTA, 1% Triton X-100, 100 mM NaCl, inhibitors), twice in the previous buffer lacking NaCl, and twice in kinase buffer (20 mM Tris pH 7.4, 10 mM $MgCl_2$, 1 mM DTT), and incubated for 40 min at room temperature in kinase buffer containing 1 μM cold ATP, 5 μCi [γ-32P]ATP (Perkin Elmer, Greenville, SC) and 1 μg of GST-CRK substrate. Kinase reactions were run on SDS-PAGE gels (10%), stained with Coomassie blue (0.25 in 50% methanol/10% glacial acetic acid), destained, dried, and exposed to film. Bands were quantitated with ImageJ64 (online tool; v1.44o). We previously documented the high specificity and sensitivity of this assay[62].

*Recombinant kinase assays*: His-tagged, recombinant ABL1 or ABL2 (ThermoFisher, Waltham, MA; 25 ng) were incubated with His-tagged, full-length MAP3K1 (contains some breakdown; Origene; 100 ng), and/or GST-14-3-3-ε (100 ng) in the above kinase assay buffer containing 1 mM cold ATP, and no "hot" ATP, and incubated for 30′ (or 15′ where indicated) at 37 °C. Kinase assays were run on SDS/PAGE gels and blotted with phospho-tyrosine antibody.

### Immunoprecipitation-IP/co-immunoprecipitation.
293T cells were transfected with His-tagged MAP3K1 in the absence/presence of constitutively active ABL1 or ABL2 (PP), using the calcium phosphate method[63], cells were lysed in RIPA buffer, lysate was precleared with NTA-agarose (Qiagen, Germantown, MD), and His-tagged MAP3K1 was pulled down with nickel agarose (Qiagen). After 3 h incubation, precipitates were washed 3× in lysis buffer containing 20 mM imidazole, resuspended in 2× SDS-sample buffer containing 400 mM imidazole, and run on SDS-PAGE gels. For coIPs, cells were lysed in TNEN buffer (50 mM Tris, pH 7.5, 50 mM NaCl, 2 mM EDTA, 0.5% NP-40 + inhibitors—see above)[16,64], lysate precleared with mouse or rabbit IgG and protein-A (rabbit) or protein G (mouse) sepharose (1 h), immunoprecipitated with agarose-conjugated antibodies (3 h), washed 3× in lysis buffer, and complexes run on SDS-PAGE gels and blotted.

### Subcellular fractionation.
Cytoplasmic/nuclear lysates were prepared with the NE-PER kit (ThermoFisher). Equal percentages of cytoplasmic and nuclear fractions were loaded for each cell line, and ABL1 cytoplasmic:nuclear ratios were then compared between cell lines following quantitation with ImageJ64 (v1.44o).

### Viability assays.
*CellTiter Glo (CTG):* Viability was assessed by plating cells in 96-well plates, treating the next day with drugs for 72 h, followed by harvest with CellTiter Glo (Promega, Madison, WI) according to the manufacturer's instructions. Cells were subconfluent at harvest. Assays were performed using three-drug doses (alone/combination). All cell lines were taken off of drugs for 2 days prior to plating cells for experiments. *MTT assays*: Cells were plated in 96-well dishes, and drug-treated. After 72 h, washed cells were treated with MTT solution (5 mg/ml; 4 h, 37 °C). The supernatant was removed, and formazan crystals were dissolved with 100 μl DMSO. Absorbance at 570 nm was measured with a microplate reader.

### Western blotting.
Equal amounts of cellular lysate (BCA protein estimation; BioRad; Hercules, CA) were run on SDS-PAGE gels, transferred to nitrocellulose, blocked with 5% milk/TBST (0.1%), and incubated with antibodies at the concentrations and stringencies noted in Supplementary Table 2.

### In vivo assays.
(1) *Mel1617 reversal of BRAFi resistance xenograft assay:* Mel1617 or Mel1617-BR ($2 \times 10^6$) cells (in 1× Hank's Balanced Salt Solution; HBSS; Lonza, Greenwood, SC) were injected subcutaneously into SCID-beige mice (6 weeks old, female) together with matrigel (100 μl high concentration, BD Biosciences 1:1 ratio with HBSS). When tumors reached a mean volume of 200 mm³, parental control mice were randomly assigned to either vehicle or PLX4720, whereas mice harboring Mel1617-BR tumors were randomly assigned to vehicle (DMSO: PEG400: Vitamin E TPGS: Poloxamer 407; PBS 1× 5:25:10:1:59), nilotinib (33 mg/kg, b.i.d.; o.g.), PLX4720 (25 mg/kg, 1× daily, o.g.) or combination treatment groups. PLX4720 dosage was 50 mg/kg/day initially, as recommended by Plexxikon Inc., but due to anorexia, the dose was reduced to 25 mg/kg after 2 days. Mice were treated until the study endpoint (largest tumors reached 1500 mm³ or were ulcerated, thus requiring euthanasia).

(2) *M14-BMR-reversal of BRAFi/MEKi resistance xenograft assay:* Parental M14 cells or BRAFi/MEKi-resistant M14-BMR cells ($5 \times 10^6$ cells) were injected subcutaneously into athymic nude mice (6 weeks old, female). When the mean tumor size was 200 mm³, mice harboring resistant cells were randomized to vehicle (see above), nilotinib (33 mg/kg, b.i.d.; o.g.), dabrafenib (25 mg/kg, 1×/daily, o.g.; D) + trametinib (0.15 mg/kg, 1× daily, o.g.; T) or the combination, whereas mice harboring parental xenografts were treated with vehicle or D/T. Parental-vehicle-treated mice were euthanized when the largest tumor approached 1500 mm³. For mice bearing M14-BMR tumors, all vehicle- and nilotinib-treated mice and four animals from D/T and D/T+ nilotinib groups were euthanized on day 35 (some vehicle and nilotinib tumors had reached 1500 mm³), and residual tumor/matrigel dissected. The remainder of D/T— and D/T+ nilotinib-treated mice (*n* = 8/group) were followed long-term, and D/T mice were euthanized when each tumor reached 1500 mm³. When some D/T+ nilotinib animals eventually relapsed (≅day 95), they were treated with ponatinib (30 mg/kg/d, o.g.). If mice showed signs of toxicity (weight loss, anorexia), ponatinib was withdrawn until symptoms improved.

(3) *In vivo resistance assay:* To mock resistance that occurs in patients, parental, BRAFi/MEKi-sensitive M14 melanoma cells were injected into athymic nude mice, and when the mean tumor volume reached 200 mm³, mice were randomly assigned to vehicle, nilotinib, D/T, and D/T + nilotinib groups. Tumors initially regressed on D/T, but later developed resistance, similar to what is observed in patients. A cell line

was obtained from a resistant tumor by excising the tumor under sterile conditions, mincing into 3–4 mm pieces, treating with collagenase B (100U/ml; Roche) and dispase (1.5U/ml; Roche) in HBSS for 20′ at 37 °C (5 ml volume), centrifuging for 5′ at 800g, resuspending the cells in media containing D/T, and plating in a 100 mm dish. Mice were euthanized when each tumor reached 1500 mm³. All animal experiments were performed in the University of Kentucky Biomedical Biological Sciences Research Building vivarium barrier facility with ambient air temperature of 70 °F, cage temperature of 75–76 °F, 50% humidity, and 14/10 light/dark cycle. All animal experiments were approved by the Institutional Care and Use Committee (IACUC) at the University of Kentucky (Protocol #00946M2005), and experiments were performed in accordance with University and NIH guidelines.

**Immunohistochemistry.** Tumors from mouse xenograft studies were fixed in formalin (24 h), followed by incubation in 70% ethanol (at 4 °C) until sectioning. The Markey Cancer Center Biospecimen and Tissue Procurement Shared Resource performed immunohistochemistry. Slides containing 4 μm sections were stained with anti-MYC antibody (Y69, Roche Catalog #790-4628) on a Ventana Discovery Ultra auto-stainer (antibody is prediluted at an estimated concentration of 24 μg/ml). Visualization was carried out with Ventana Ultramap Alkaline Phosphatase reagent (Roche # 760-4314) and Discovery Red Chromogen (Roche #760-228) according to the manufacturer's recommendations. Slides were blindly scored by a pathologist (D. Richards), and photographs were obtained following scanning on an AperioScan Digital Scope (Leica Biosystems, Cincinnati, OH).

**Statistical analysis.** *General:* For in vitro studies, two-sample (comparisons between-treatment groups) or one-sample *t*-tests (comparisons against normalized controls) using the Holm's method for multiple comparisons adjustment was performed. For in vivo studies, ANOVA with Sidak correction for multiple comparisons was used to compare tumor volume across treatment groups. Kaplan–Meier curves and the logrank test with the Holm's method for multiple comparisons adjustment were used to compare time-to-tumor doubling across treatment groups. For all analyses, two-tailed *P* values are reported.

*Analysis of Existing Microarray and mRNA-Seq Clinical Datasets (Pre-/Post-ERK pathway therapy):* Datasets for Fig. 1d/Supplementary Fig. 1f were downloaded from the GEO database (accession #: 50535, 50509, 77940), and bam files for EGAS1000992 were kindly provided by Drs. Lawrence Kwong (MD Anderson, Houston, TX). A two-sided binomial was utilized to analyze whether *ABL1* mRNA was elevated or downregulated following resistance. In Fig. 6b, three mRNA-Seq datasets were utilized: GSE50535, GSE77940, and EGAS00001000992. Since the sample sizes of these datasets were limited, in order to increase sample size and therefore statistical power, we combined datasets with the R package "sva", which removes batch effects using an empirical Bayesian framework[65]. After combining the datasets, Spearman's rank-order correlation analysis was performed to test the correlations between *ABL1* and *MYC* before and after treatment, respectively. For Supplementary Fig. 5c, RNASeq data were downloaded from Genomic Data Commons (GDC) for TCGA Skin Cutaneous Melanoma data (access date, November 2019) and normalized to Transcripts Per Kilobase Million (TPM). Spearman's correlation coefficients were used to quantify the correlation between *ABL1* and *MYC* mRNAs.

**Sequencing.** *Whole-exome sequencing:* WES was performed by the University of Kentucky Markey Cancer Center Oncogenomics Shared Resource. DNA was sheared using the Covaris E220 sonicator. Library preparation and exome capture was performed using the Agilent SureSelectXT Clinical Research Exome v2. Paired-end sequencing was done on an Illumina HiSeq 2500 in rapid run mode (2 × 100 bp). Sequencing reads were trimmed and filtered using Trimmomatic (V0.39)[66], then aligned to human reference genome b37/hg19 using BWA (V. 0.7.17)[67]. PCR duplicates were removed using Picard (v2.20.2; http://broadinstitute.github.io/picard/). The Genome Analysis Toolkit (GATK v4.1.2.0)[68] was used for base quality score recalibration. For comparisons between resistant lines and their parental counterparts, somatic mutations present in the resistant line but not in the parental line were detected using Mutect2[69] (part of the GATK toolkit) tumor/normal mode with the parental line as the reference. Mutations passing the contamination filter and orientation bias filter were annotated (Oncotator, v1.9.9.0)[70].

Sanger sequencing of *MAP2K1* in genomic DNA isolated from 451-Lu-BR cells was performed by ACGT, Inc. using the primers described in Supplementary Table 3, and visualized with 4 Peaks software (free online tool).

**Reporting summary.** Further information on research design is available in the Nature Research Reporting Summary linked to this article.

## Data availability

Next-generation sequencing (WES) data have been deposited in the NCBI/SRA database under the accession code (PRJNA639552). Links to data analyzed from the GEO database (accession #s 50509, 5077940, 50535, 65185), Genomic Data Commons (GDC), and link to the official database housing EGAS1000992 (data kindly provided by Dr. Lawrence

Kwong; MD Anderson, TX; European Genome-phenome Archive) are below. https://portal.gdc.cancer.gov/projects/TCGA-SKCM; https://www.ebi.ac.uk/ega/datasets.

All the other data supporting the findings of this study are available within the article and its supplementary information files and from the corresponding author upon reasonable request. Source data are provided with this paper.

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

## Acknowledgements

This work was supported by the Markey Cancer Center Biospecimen and Tissue Procurement, Flow Cytometry, Oncogenomics, and Biostatistics and Bioinformatics Shared Resources, as well as the Research Communication Office (P30CA177558). We thank Dr. Lawrence Kwong (MD Anderson) for EGAS RNAseq data; Novartis for supplying nilotinib and dabrafenib; Plexxikon Inc. for supplying PLX4720; Dr. Meenhard Herlyn (Wistar Institute) for supplying 451-Lu, Mel1617, and WM164 cell lines; and Drs. Qing-Bai She and Vivek Rangnekar (University of Kentucky) for critically reading the manuscript.

## Author contributions

A.J. contributed CellTiter Glo data (one biological replicate) in Fig. 2a, b and Supplementary Fig. 2a, b; performed Supplementary Fig. 10b, as well as a few biological replicates for blots shown in graphs for Fig. 4a, b. She also established the M14-BR cell line. Z.L. performed some western blots for Figs. 2g, 4b, c and 5c and Supplementary Figs. 4a, 5b, 6a, b, d, g, made some lysates for those western blots, and performed/assisted with tumor cell injections and tumor removal (in vivo). A.L. performed western blots in Supplementary Fig. 5a and colony assays in Fig. 2f. C.M. performed colony assays in Supplementary Fig. 3. D.R. read/analyzed immunohistochemistry (Fig. 8e and Supplementary Fig. 8b, c). C.W., J.L., and D.H. performed statistical analyses as follows. Figures 1d, 6b, Supplementary Figs. 1f, 5c, 6b (D.H.), Supplementary Table 1, Fig. 8 (C.W.), and WES (J.L.). M.N. and A.R. developed the vehicle formulation for PLX4720 + nilotinib in vivo studies and provided advice for the studies. Melanoma oncologists, M.W. and P.W., provided advice for in vivo studies. R.P. performed experiments in Figs. 1a, e, 7, and Supplementary Figs. 1g, 6j, and 7; made the *ABL1/2* shRNA and ERK2-GOF-expressing lines; performed two- and one-sample *t*-tests; directed the project; and wrote

the paper. R.T. performed all other experiments and contributed lysate replicates for most western blot figures.

## Competing interests

This research is/was funded by the following grants to R.P.: National Institute of Health (NCI): R01 CA211137 and R01 CA166499; Lloyd Charitable Trust; University of Kentucky Markey Foundation Women's Strong Award; and Cancer Center Support Grant Pilot Award (5P30CA177558). M.N. and A.R. are employees of Plexxikon Inc.; all other authors declare no competing interests.
