## [Peer Review File · Nature Communications]

Reviewers' comments:

Reviewer #1 (Remarks to the Author); expert in melanoma resistance, mouse models:

The manuscript by Tripathi, Plattner and colleagues investigates the effect of targeting ABL1/2 mediated reactivation of MEK pathways in ablating acquired BRAF/MEKi resistance in melanoma. This is of great clinical significance given the alarming rate of non-responders and patients who quickly gain resistance to BRAF/MEKi.

Overall the manuscript is well written and the experiments are set out in a logical manner. The authors should be praised for their elegant use of whole genome sequencing across various resistance stages and this will be of great use to the field. Furthermore, the authors go into great depth in terms of investigating both single agent resistance and combo. Figure 7 shows great promise in a pre-clinical setting and overall should provide great interest to the field. The methods have appropriate detail to repeat the experiments performed and the statistical analysis appears appropriate.

There are a few technical issues that I feel need to be addressed before the manuscript is ready for submission. In particular, many of the western blots shown early on have very unequal loading controls displayed, some of which are well over 2-4 fold in difference. While densitometry analysis has been performed in these cases, some of the loading is so dramatically different that analysis by densitometry would not be accurate. Given that many of the western blots are said to have been performed more than N=3 in the figure legends, at the very least, representative blots need to be displayed with roughly equal loading. This is required for the following blots:

Figure 1A: the M14 and M14 BR is dramatically lower than the BMR sample (well over 2 fold). The 451-LU is also dramatically lower than the 451-LUBR. In both cases, the actin follows the trend they describe, thus these blots need to be repeated with a more equal loading. Furthermore, you mention the stats for this in the figure legend. I feel it would be better presented in bar graphs in a sup, it is too confusing in the figure legend (like you have done for figure 4).

Figure 1B: The tubulin between M14, M14 BR and M14BMR cyto is not acceptable (BR vs BMR is well over 4 fold different.) 451 LU vs luBR is also very low. Needs to be repeated and a proper representative image displayed.

Figure 1C needs to be written clearer in the results to indicate it is quantification of all three lines combined.

Figure 1E: Loading control is not equal, BMR is dramatically different to the other lines.

Other minor comments that need to be addressed:

Introduction

Many citations in the introduction would be easier to follow when placed after a statement has been made rather than adding them all at the end of longer sentences. The major one would be the sentence starting with Tumors circumvent the effects...

You mention in the discussion that the toxicity of nilotinib is well known. You should briefly mention the toxicity of this drug in the clinic in the introduction.

Results

A brief one sentence description of the kinase assay should be done in the results section when first mentioned in figure 1.

Figure 6D MAP3k8 already looks reduced in the D/T treatment compared to the parental and also compared to the vehicle resistant. I recommend taking these blots and the sentence describing this out. It doesn't add to the story.

Figure 6G: Does this correlation stand in normal melanoma tcga? It would be interesting to also show this using the TCGA dataset given the sample size is so much higher.

Reviewer #2 (Remarks to the Author); expert in ABL signalling and therapy:

Tripathi and colleagues demonstrate using a panel of sensitive and resistant melanoma cell lines a critical role of the ABL1/2 tyrosine kinases in melanoma resistance to BRAF/MEK inhibitor combinations. In particular, the use of the approved BCR-ABL inhibitor nilotinib reverses resistance in cell lines and in mouse xenograft models.

Overall, this is an interesting manuscript that reports a novel resistance mechanism in melanoma and provides strong evidence for the clinical evaluation of nilotinib in melanoma patients that are resistant to standard-of-care BRAF/MEK inhibitor combinations. The manuscript provides a large amount of data that is overall well controlled. Still, there are a number of points listed below that should be experimentally addressed and/or clarified.

1. Data should be included in the manuscript demonstrating that the 'resistant' cell lines (e.g. M14-BMR and Mel1617-BMR) are indeed resistant (and how strong the resistance is) to BRAF/MEK inhibitors as compared to the parental cells, e.g. by measuring IC50 curves using MTT/XTT, CellTiter Glo are another common cell growth read-out.
2. Fig. 1a (and 1e) shows panels for ABL1 and ABL2 kinase assays, respectively. The M&M section describes a standard ABL IP-kinase assay using the ABL1 antibody K12 (which as far as I know does not IP ABL2) and using GST-CRK as a substrate. How can one distinguish ABL1 from ABL2 kinase activity then?
3. Fig 1b: In the upper panel (M14 cells), beta-tubulin levels are extremely different between sensitive and resistant cell lines, which strongly compromises the conclusion on different cyto:nuclear ratios. In the two panels below, loading/fractionation control levels are more equal. Experiment for the M14 cells should be repeated.
4. Fig. 1d: Can the authors provide a number of the fold-change in ABL mRNA levels that is observed in the patient samples? What was the minimum fold-change from which up- or down-regulation was called?
5. In Fig. 1E, SU6656 is used at 10µM for 16h. At this concentration SU6656 directly inhibits ABL1/2 (IC50 of SU6656 for ABL1/2 is ~800nM) in addition to SFKs. Experiment needs to be repeated with lower SU6656 concentrations (e.g. 500nM) to be able to support the conclusion that 'SRC activation contributes to ABL1/2 activation during resistance'.
6. Fig. 2h: A control immunoblot showing the expression levels of ABL1 PP and ABL2 PP as compared to endogenous ABL1/2 need to be shown to support the conclusion.
7. The authors should add a cautionary statement to the manuscript, that not all effects that they observed with nilotinib are due to on-target inhibition of ABL1/2, but may be due to off-target inhibition of BRAF V600E (and other MAPK) by nilotinib. As demonstrated by Manley et al. (2010) BBA, 1804(3), 445-453, and others, nilotinib has a variety of off-targets that are inhibited in the concentration range (2-6 µM) that the authors use for all of their experiments. Instead of nilotinib, asciminib (ABL001), which is the cleanest ABL1/2 inhibitor available, should be used at low nM concentrations. GNF5 is outdated, binds with extremely low affinity, and lacks clinical relevance.
8. In Fig. 6, data on the resistance mechanism is presented and the role of ABL1/2 downstream of InsR/IGF1R and upstream of different MAP3Ks is described. But the data shows mere correlations and I am confused about the underlying molecular mechanisms: Are MAP3Ks substrates of ABL1/2? Where are they phosphorylated and what is the consequence on MAP3K activity of these

phosphorylation events? Or does ABL1/2 activity alter protein expression levels of MAP3Ks and if so how?

9. The authors mention at the end of the discussion section a planned phase 1 clinical trial with the triple combination. Following up on one of my points above, the authors should comment on the known plasma levels of nilotinib in CML patients compared to the concentrations used on the cell line and mouse models in this study: Are the nilotinib concentrations used in this study clinically achievable with the approved nilotinib doses?

Reviewer #3 (Remarks to the Author); expert in melanoma:

In this manuscript, the authors evaluate the role of ABL1/2 kinases in resistance to BRAF inhibitor therapy. They find that ABL1/2 mRNA and activity are increased in resistant patients, which is associated with activation of the MAPK pathway. Finally, the authors find that nilotinib can reverse resistance to BRAFi, or prevent de novo resistance, suggesting that it may improve the management of patients with BRAF mutant melanomas.

1. The major issue is novelty. Numerous other manuscripts (dating back to 2015) have demonstrated that SRC family kinases can overcome resistance to BRAF inhibitors. This manuscript is consistent with these reports by adding Abl as a mechanism acting through this same pathway.
2. Many experiments are lacking controls. For example:
 - a. It is not clear how specific the ABL kinase assays used are. Demonstrating that suppression of activity upon the specific knockdown of the respective kinases would be necessary to confirm specificity. Alternatively, demonstration of increased Abl phosphorylation, which is required for its activity would be an adequate orthogonal way to confirm the conclusions.
 - b. shRNA experiments need a cDNA rescue to confirm specificity
 - c. Drug experiments need a drug resistant mutant, not a gain of function mutant as they authors use.
3. In many blots and other experiments, it appears that the level of ABL protein are increased in the resistant cells. In fact the authors argue that ABL is increased in resistant cells at the mRNA level (Figure 1d), which suggest that the mechanism by which ABL1 is activated is not exclusively related to upstream signaling. Is there any evidence of increased activity of ABL kinases in resistant tumors?
4. Are the authors suggesting that ABL1 is dysregulated but not ABL2, as they present only the mRNA for ABL1 (Figure 1d)?
5. The data presented in Figure 2 seems to suggest that nilotinib has similar effects in BRAF-sensitive and BRAF-resistant clones (Figure 2a, Figure 2c, Figure 2e), which do not fit with the model suggested by the authors that ABL is activated upon the development of BRAFi resistance

Responses to Referees

We thank the reviewers for their careful and thorough examination of our manuscript. We were pleased that the reviewers recognized that the manuscript described data that were of “great clinical significance”, was “well-written”, detailed a “novel mechanism of resistance” and included “a large amount of data” that were “overall, well-controlled”. The reviewers had a few concerns, which we have addressed in full below, as well as within the manuscript (significant changes in the text are denoted with yellow highlighting).

Reviewer #1.

1. “Figure 1A: The M14 and M14 BR (actin) is dramatically lower than the BMR sample...“Figure 1E: Loading control is not equal, BMR is dramatically different to the other lines.” We have performed many western blots with these lines, and we noticed that M14-BMR cells express different levels of structural proteins, such as actin, than parental or -BR cells, which likely is due to the large, flat morphology of BMR cells. These differences appear to be accentuated when cells are in serum-starved conditions (Fig. 1; necessary when determining basal activities) as opposed to when they are grown in serum conditions (Fig. 4,5). We performed a BCA and loaded equal protein, and the loading indeed appears even by ponceau staining. We blotted the lysates in question with other loading controls (beta-tubulin, GAPDH) and found that the levels are more even (or even underloaded in the case of M14-BMR), indicating that actin levels are indeed different between lines. Thus, we used a different loading control for Fig. 1a,e, and Supplementary Fig. 1g.

2. “Figure1B: The tubulin between M14, M14 BR and M14BMR cyto is not acceptable (BR vs BMR is well over 4 fold different)...Needs to be repeated and a proper representative image displayed.” In this experiment, we are comparing ABL1 levels in the cytoplasm to ABL1 levels in the nucleus for each line in order to obtain a cytoplasmic:nuclear ratio. To get this ratio, we must load an equal fraction of cytoplasmic and nuclear lysate for each line (e.g. 1/20 of the total of each fraction). Once we obtain a cyto:nuc ratio for each line, we compare the ratios between lines. Thus, it really isn’t imperative that loading be exactly the same between lines as we are never comparing ABL levels between lines. The tubulin and lamin blots were included to demonstrate that the fractions are pure (no contamination of cytoplasmic proteins in the nuclear fractions and vice-versa). However, in the revised manuscript, we performed a BCA protein estimation on cytoplasmic fractions in order to try to load similar levels of cytoplasmic fractions between lines. However, we found that despite loading the same amount of cytoplasmic protein (ponceau levels were equivalent), tubulin and lamin levels were not equivalent between the M14 cell lines as the lines express different amounts of these structural proteins. Indeed, when we artificially altered the loading of the lines in order to try to obtain nearly equivalent tubulin levels across lines (Fig. 1b), lamin levels still were low in M14-BR cells, indicating that lamin expression is not equivalent among the lines. We cannot artificially alter the nuclear and cytoplasmic protein levels to keep them even among the lines, as it is imperative to load equal fractions of cytoplasm vs. nucleus within each line in order to obtain accurate cytoplasmic:nuclear ABL ratios. Importantly, as expected, even though we artificially changed the loading, since we still loaded equivalent cytoplasmic:nuclear fractions within each line, the result remains the same (Fig. 1b). We attempted to clarify these points within the current version of the manuscript. Finally, we also now provide additional evidence to corroborate our findings by demonstrating that ABL1 is more highly phosphorylated on T735 (14-3-3 binding site)^{1,2} in BRAFi/MEKi resistant cells. Phosphorylation of this site is known to induce cytoplasmic retention of ABL1, which drives its oncogenic function (Fig. 1c).

3. “Figure1C needs to be written clearer in the results to indicate it is quantification of all three lines combined.” This was done—please see page 5, and **Supplementary Fig. 1e** legend.

4. “Furthermore, you mention the stats for this (Fig. 1) in the figure legend. I feel it would be better presented in bar graphs in a sup, it is too confusing in the figure legend (like you have done for figure 4).” Quantitation for this figure was moved to **Supplementary Fig. 1d**.

5. “Many citations in the introduction would be easier to follow when placed after a statement has been made rather than adding them all at the end of longer sentences...” As suggested by the reviewer, the references were separated.

6. “...the toxicity of nilotinib is well known. You should briefly mention the toxicity of this drug in the clinic in the introduction.” We added this statement on pages 3/4.

7. “A brief one sentence description of the kinase assay should be done in the results section when first mentioned in figure 1.” We added this clarification on page 5.

8. “Figure 6D MAP3k8 already looks reduced in the D/T treatment... recommend taking these blots and the sentence describing this out...” We removed the MAP3K8 blot from new Fig. 6c, and the sentence that describes it from the manuscript.

9. “Figure 6G: Does this correlation stand in normal melanoma tcga? It would be interesting to also show this using the TCGA dataset given the sample size is so much higher.” The TCGA dataset does not include post-resistance samples (unlike the datasets we queried), which is why this dataset was not used, initially. However, since we also observed a positive correlation between *ABL1* and *MYC* prior to resistance, we examined the correlation in the TCGA dataset. Indeed, we found *ABL1* and *MYC* mRNAs were highly correlated in this dataset as well (see **Supplementary Fig. 5c**; $p=2.29E-5$).

Reviewer #2.

1. “Data should be included in the manuscript demonstrating that the 'resistant' cell lines (e.g. M14-BMR and Mel1617-BMR) are indeed resistant (and how strong the resistance is) to BRAF/MEK inhibitors as compared to the parental cells...” These data are now included in **Supplementary Fig. 1a-c**. We show all lines that were not previously published (M14-BR, M14-BMR, Mel1617-BMR).

2. “Fig. 1a (and 1e) shows panels for ABL1 and ABL2 kinase assays, respectively. The M&M section describes a standard ABL IP-kinase assay using the ABL1 antibody K12 (which as far as I know does not IP ABL2) and using GST-CRK as a substrate. How can one distinguish ABL1 from ABL2 kinase activity then?” We apologize for the confusion and the mistake in the “Materials and Methods”. We immunoprecipitated ABL1 with K12 antibody, and ABL2 with an ABL2-specific antibody that we previously made/characterized.^{3,4} ABL1 and ABL2 IPs were incubated with GST-CRK substrate and radiolabeled (γ -³²P) ATP. This has been clarified in the “Results”, “Materials and Methods” and “Figure Legends”.

3. “Fig 1b: In the upper panel (M14 cells), beta-tubulin levels are extremely different...” In this experiment, we are comparing ABL1 levels in the cytoplasm to ABL1 levels in the nucleus for each line in order to obtain a cytoplasmic:nuclear ratio. To get this ratio, we must load an equal fraction of cytoplasmic and nuclear lysate for each line (e.g. 1/20 of the total of each fraction). Once we obtain a cyto:nuc ratio for each line, we compare the ratios between lines. Thus, it really isn't imperative that loading be exactly the same between lines as we are never comparing ABL levels between lines. The tubulin and lamin blots were included to demonstrate that the fractions are pure (no contamination of cytoplasmic proteins in the nuclear fractions and vice-versa). However, in the revised manuscript, we performed a BCA protein estimation on cytoplasmic fractions in order to try to load similar levels of cytoplasmic fractions between lines. However, we found that despite loading the same amount of cytoplasmic protein (ponceau levels were equivalent), tubulin and lamin levels were not equivalent between the M14 cell lines as the lines express different amounts of these structural proteins. Indeed, when we artificially altered the loading of the lines in order to try to obtain nearly equivalent tubulin levels across lines (Fig. 1b), lamin levels still were low in M14-BR cells, indicating that lamin expression is not equivalent among the lines. We cannot artificially alter the nuclear and cytoplasmic protein levels to keep them even among the lines, as it is imperative to load equal fractions of cytoplasm vs. nucleus within each line in order to obtain accurate cytoplasmic:nuclear ABL ratios. Importantly, as expected, even though we artificially changed the

loading, since we still loaded equivalent cytoplasmic:nuclear fractions within each line, the result remains the same (Fig. 1b). We attempted to clarify these points within the current version of the manuscript. Finally, we also now provide additional evidence to corroborate our findings by demonstrating that ABL1 is more highly phosphorylated on T735, (14-3-3 binding site)^{1,2} in BRAFi/MEKi resistant cells. Phosphorylation of this site is known to induce retention of ABL1 in the cytoplasm which drives its oncogenic function (Fig. 1c).

4. “Fig. 1d: Can the authors provide a number of the fold-change in ABL mRNA levels that is observed in the patient samples? What was the minimum fold-change from which up- or down-regulation was called?”

Due to the small number of matched (pre-/post-treatment) samples that were available, all changes were included in the graph. However, we now also include additional figures where we present changes greater than particular cut-off values (**Supplementary Fig. 1f**).

5. “In Fig. 1E, SU6656 is used at 10microM for 16h. At this concentration SU6656 directly inhibits ABL1/2 (IC50 of SU6656 for ABL1/2 is ~800nM) in addition to SFKs. Experiment needs to be repeated with lower SU6656 concentrations (e.g. 500nM)...”

We could not find the reference to which the reviewer is referring. According to the initial description of SU6656, it displays much lower affinity towards ABL than SRC (*in vitro*, ABL IC₅₀=1.74μM, SRC IC₅₀=0.28μM),⁵ unlike other SRC kinase inhibitors that target ABL and SFKs equally (e.g. dasatinib, bosutinib). It is not possible to estimate the SU6656 IC₅₀ for ABL1/2 in cells unless all expressed SFKs are absent since SFKs directly phosphorylate and activate ABL1/2.⁶ Indeed, we showed in our “*Cancer Research*” paper in 2006, that SU6656 treatment (10μM) of knockout MEFs lacking all three SFKs expressed in skin (Src, Fyn, Yes) has no effect on ABL1 or ABL2 kinase activity, whereas the same dose of SU6656 reduces the activities of ABL1 and ABL2 in breast cancer cells expressing activated SFKs (**Appendix Fig. 1a**). Thus, SU6656 does not directly inhibit ABL1/2 at this dose but instead inhibits ABL1/2 by inhibiting SFK-activation of ABL1/2. We used 10μM SU6656 for our experiments because in the original paper describing SU6656,⁵ 10μM was required to efficiently inhibit SFK activity in cells, and we found the same to be true in melanoma cells, as lower doses of SU6656 did not efficiently inhibit SFKs in our lines (**Appendix Fig. 1b,c**). Indeed, even 10μM SU6656 only partially inhibited SFKs (see phospho-SRC; **Fig. 1e, Supplementary Fig. 1g**), as it blocked SFK activity 30-50% in Mel1617 cell lines (parental=30%; BR=45%), and 25-65% in M14 lines (parental=25%, BR=19%, BMR=65%). Finally, if 10μM SU6656 directly inhibits ABL1 and ABL2 as postulated by the reviewer, one would expect this concentration to inhibit ABL1 and ABL2 in all lines. However, although 10μM SU6656 dramatically decreases ABL1 and ABL2 activity in M14-BR and M14-BMR, it does not inhibit ABL1 or ABL2 in Mel1617-BR; it only inhibits ABL1 (but not ABL2) in Mel1617-BMR; and it only inhibits ABL2 (but not ABL1) in 451-Lu-BR (**Supplementary Fig. 1g**), which indicates that the effect of SU6656 on ABL1/2 activity is due to the differential contribution of SFKs to ABL1/2 activation rather than due to direct effects on ABL1 and ABL2.

6. “Fig. 2h: A control immunoblot showing the expression levels of ABL1 PP and ABL2 PP as compared to endogenous ABL1/2 need to be shown to support the conclusion”. These blots were added to Figure 2h.

7. “The authors should add a cautionary statement to the manuscript, that not all effects that they observed with nilotinib are due to on-target inhibition of ABL1/2... asciminib (ABL001), which is the cleanest ABL1/2 inhibitor available, should be used...”

The indicated cautionary statement has been added to the discussion (pg. 20), although it should be noted that nilotinib’s effects are mimicked by ABL1/2-shRNA and also are consistent with gain-of-function experiments, and results with GNF-5 (discussed below). Moreover, we now show rescue experiments with T315I mutant forms of ABL1/2 that are resistant to nilotinib (**Supplementary Fig. 2g**), thereby providing additional evidence that ABL1/2 are the targets of nilotinib. Regarding ABL001, this drug was designed to inhibit BCR-ABL, and it efficiently does so.⁷ However, unfortunately, ABL001 (obtained directly from Novartis) is inefficient at inhibiting endogenous ABL1 or ABL2 in human melanoma cells, and only inhibits ABL1/2 activity by 40-60% (**Appendix, Fig. 2a,b**). In our work studying the ABL kinases over the past two decades, we’ve found that if the kinases are not efficiently inhibited (>85%), they retain the ability to signal. Indeed, ABL1/2

shRNA, GNF-5, nilotinib, and ponatinib all efficiently reverse BRAFi or BRAFi/MEKi resistance, whereas ABL001 has no effect (**Appendix, Fig. 2c,d**). Consistent with our data, in the initial report describing ABL001,⁷ the authors indicate that the drug acts specifically in cells that express BCR-ABL and has little/no activity in non-BCR-ABL-expressing cell lines including solid tumor lines known to harbor high ABL1/2 activity such as RKO-colon,⁸ A549-lung,⁹ H460-lung,¹⁰ H1915-lung,¹¹ LOX-IVMI-melanoma,¹² MDA-MB-231-breast,¹²⁻¹⁷ BT-549-breast,¹²⁻¹⁷ MDA-MB-468-breast,¹²⁻¹⁷ SKBR3-breast,¹²⁻¹⁷ etc. (see Source File from the manuscript).⁷ In contrast, numerous groups have used GNF-5 for *in vitro* and *in vivo* studies using solid tumor cell lines.^{10,18-20} Thus, although GNF-5 might not be clinically relevant, and doesn't inhibit ABL1/2 as efficiently as nilotinib, it is specific and thus, we believe our experiments using GNF-5 provide important corroborating evidence for nilotinib, shRNA/shRNA-rescue, gain-of-function, and T315I rescue studies.

8. “In Fig. 6, data on the resistance mechanism is presented and the role of ABL1/2 downstream of InsR/IGF1R and upstream of different MAP3Ks is described. But the data shows mere correlations and I am confused about the underlying molecular mechanisms...” We apologize that Fig. 6 was confusing. We believe that this was due, in part, to various cell lines harboring different ABL-dependent mechanisms of resistance. In order to reduce confusion and to address the excellent comment made by this reviewer, we focused our attention on unraveling the mechanism by which ABL kinases induce MEK/ERK reactivation, specifically in M14-BMR cells, since these cells represent a clinically relevant model due to their resistance to both BRAFi and MEKi, a current treatment regimen. In the revised manuscript, we performed a series of new experiments (summarized below; **Figs. 6 and 7 and Supplementary Figs. 6 and 7**), which provide new mechanistic insight into how ABL1/2 drive resistance. First, we demonstrate the novel finding that the MAP4K1→MAP3K1 pathway, which usually activates JNK/JUN, shifts to reactivating MEK/ERK during melanoma resistance. Moreover, we show that ABL kinases play a critical role in this pathway. ABL1/2 are in a complex with MAP4K1/MAP3K1, MEK/ERK, and 14-3-3 scaffold proteins, and their kinase activities and 14-3-3 binding are required for complex formation. Indeed, ABL1/2 phosphorylate MAP3K1 and 14-3-3, and drive coupling of MAP4K1 to MAP3K1. Since numerous MAP4Ks can phosphorylate MAP3K1, and MAP3K1 can be activated by many MAP4Ks, these data are significant as they demonstrate that ABL1/2 not only drive signaling of MAP3K1 to MEK/ERK but also mediate coupling of a specific MAP4K (MAP4K1) to MAP3K1. We thank the reviewer for this excellent suggestion which has led to the identification of novel mechanisms into how ABL1/2 drive melanoma resistance. Delineation of these novel findings has further strengthened our revised manuscript.

9. “The authors mention ... a planned phase 1 clinical trial ..the authors should comment on (whether) the nilotinib concentrations used in this study (are) clinically achievable with the approved nilotinib doses?” Indeed, we submitted a clinical trial “Concept” to Novartis which was US-approved and more recently, now, globally-approved for Phase I clinical trial initiation (Bernard Hsu, Desiree Hiramien, personal communication). We sent the IIT protocol to Novartis in March and are awaiting approval to begin accruing patients, hopefully, this year. Nilotinib is FDA-approved for drug-resistant CML at 400mg/BID dose. The 400mg dose translates to mean plasma trough levels of 1.95µM,²¹ which is within the range of concentrations that we utilized for our *in vitro* studies. Additionally, 600mg/2X daily also has been used successfully in the clinic.²² More importantly, the dose we used in our animal studies was recommended by Novartis and is the same dose they utilized (30mg/kg) for CML models. Moreover, other groups found the 20mg/kg dose is subcurative in CML models whereas 50mg/kg is curative,²³ and more recent data performed by Novartis used nilotinib at a dose of 75mg/kg in mice.⁷ Thus, the concentrations of the drug we are utilizing can be achieved in the clinic, and Novartis approved our proposal based on the data outlined in this manuscript, indicating that they also believe that an effective dose can be achieved in people. These points were added to the discussion (see pg. 22).

Reviewer #3.

1. “The major issue is novelty. Numerous other manuscripts ... have demonstrated that SRC family kinases can overcome resistance...This manuscript is consistent with these reports” We respectfully disagree with the reviewer, as we believe our manuscript is exceedingly novel for the reasons

indicated below. Although SRC can contribute to ABL activation, it isn't the entire story as inhibition of SRC only partially reduces ABL1/2 activity; SU6656 inhibits ABL1/2 in some lines (M14-BR, M14-BMR) but not in others (Mel-BR); and SU6656 inhibits ABL1 but not ABL2 in Mel-BMR, whereas in 451-BR, SU6656 inhibits ABL2 and not ABL1 (**Fig. 1e, Supplementary Fig. 1g**). Thus, other tyrosine kinases are required for full activation of ABL1 and ABL2 in some lines. Moreover, we previously showed that ABL1 upregulation at the mRNA and protein level is required for activation and occurs prior to subsequent phosphorylation by tyrosine kinase kinases, indicating that SRC activation alone isn't sufficient to activate ABL1/2 in melanoma.²⁴

We believe that our data are novel for the following reasons. 1) ABL1/2 have not previously been identified as mediators of melanoma resistance within either SRC-dependent or SRC-independent pathways. Our report is the first to link ABL1/2 to melanoma resistance; 2) We are the first to show exciting *in vivo* data demonstrating that targeting ABL1/2 prevents and reverses resistance, which provides the rationale for a clinical trial to repurpose an FDA-approved ABL1/2 inhibitor (which does not inhibit SRC) for treating treatment-refractory melanoma. Indeed, in collaboration with Novartis, we developed an IIT and will enroll patients at two sites (University of Kentucky, Thomas Jefferson Kimmel Cancer Center) in the near future. 3) We identify a new mechanism by which ABL1/2 drive resistance, demonstrating that they drive reactivation of ERK by impacting MAP3K1, a mechanism that has not been documented for SRC. We also show that ABL1/2 are in a complex with MAP4K1, MAP3K1, MEK, and ERK, and are critical for coupling MAP4K1 to MAP3K1 and subsequent MEK/ERK reactivation during resistance. 4) We are the first to demonstrate that MAP4K1 activation of MAP3K1 induces MEK/ERK reactivation during melanoma resistance. 5) All of the previous data implicating SRC in resistance have focused on its role in BRAFi resistance. BRAFi/MEKi resistance is now more relevant, given that the combination rather than BRAFi alone is standard-of-care, and BRAFi/MEKi resistance is driven by different mechanisms. Here, we show that ABL1/2 drive ERK-dependent as well as ERK-independent BRAFi/MEKi resistance, which has never been shown before for either SRC or ABL1/2.

2a. “It is not clear how specific the ABL kinase assays used are ...” demonstration of increased Abl phosphorylation, which is required for its activity would be an adequate orthogonal way to confirm the conclusions.” Previously, we described, in great detail, the high degree of specificity of the ABL1/2 kinase assay and included many different controls (see Supplementary data from the publication).⁶ Moreover, the ABL2 antibody is also described in Finn et al. *Nature Neuroscience* 6(7)717-723, 2003. This assay has subsequently been used not only by our laboratory but also by many other investigators. The kinase assay is not only more specific but also exceedingly more sensitive than the phospho-specific antibodies that have been developed against the two residues that reflect ABL1/2 activity (Y412/Y245; in ABL1; Y439/Y272 in ABL2). These antibodies are so weak that they can only accurately measure activation of BCR-ABL, and it is very difficult (if not impossible) to use them to measure activation of endogenous ABL1/2. Moreover, due to the high degree of homology between ABL1 and ABL2 kinase domains, these antibodies bind to both ABL1 and ABL2, and thus, unlike the kinase assay, cannot be used to measure ABL1 and ABL2 independently. Because of these issues, most laboratories that study ABL1/2 in solid tumors either utilize kinase assays to directly measure ABL1 and ABL2 activities or measure phosphorylation of their substrates, CRK and CRKL, on ABL1/2 phosphorylation sites (the pCRKL antibody reacts with pCRK), which is much more sensitive than the phospho-specific antibodies, and has been extensively characterized as an accurate readout of ABL1+2 activities.^{15,25} Consistent with our kinase assays, pCRKL blots demonstrate increased ABL1+2 activity in resistant cells (parental vs. resistant; DMSO-treated; **Figs. 1e, 4,5, Supplementary 1g**).

2b. “shRNA experiment needs a cDNA rescue to confirm specificity.” We performed the indicated experiment, and show that re-expression of *ABL1/2* indeed rescues *ABL1/2* shRNA-mediated inhibition of survival (**Supplementary Fig. 3d**), confirming the specificity of the shRNA, and providing corroboration with other studies using this shRNA sequence.²⁶

2c. “Drug experiments need a drug resistant mutant...” We expressed nilotinib-resistant mutant forms of ABL1/2 (T315I),²⁷ and demonstrate that this rescues nilotinib-mediated inhibition of survival (**Supplementary Fig. 2g**), indicating that nilotinib’s effects are indeed mediated by ABL1/2.

3. “In many blots and other experiments, it appears that the level of ABL protein are increased in the resistant cells. In fact, the authors argue that ABL is increased in resistant cells at the mRNA level (Figure 1d), which suggest that the mechanism by which ABL1 is activated is not exclusively related to upstream signaling. Is there any evidence of increased activity of ABL kinases in resistant tumors?” Previously, we found that *ABL1* and/or *ABL2* mRNA levels are elevated in primary melanomas, and also are upregulated in metastases as compared to the primary site.¹³ Moreover, we found that mRNA/protein upregulation is an early necessary but not sufficient step in ABL1 activation in melanoma, and occurs prior to subsequent phosphorylation/activation by tyrosine kinases.^{24,28} Thus, mRNA levels correlate with resistance or progression if activating tyrosine kinases are also present in the samples.

To determine whether ABL kinase activities are potentiated following resistance, we aimed to collect patient samples pre- and post- MAPKi treatment and stain samples with pCRKL antibody (readout of ABL1/2 activity).¹⁵ In collaboration with the Cancer Prevention and Control Program at the University of Kentucky using the Kentucky SEER registry, we went through the charts of all patients within the state of Kentucky who were diagnosed with metastatic melanoma and treated with MAPKi from 2010-2017 in order to identify patients who had matched samples prior to and after MAPKi treatment. After evaluation of the charts of >800 patients, we were unable to find any cases that met our criteria (pre-/post-samples were available and patients were treated only with a MAPKi). Indeed, post-treatment samples did not exist for nearly all patients as it was not standard-of-care to take biopsies post-progression. Post-treatment samples were available for three patients; however, the patients had been treated with numerous other therapies (e.g. radiation, immunotherapy, etc.) prior to sample collection. We attempted to obtain samples that met our criteria from other institutions that had collected the samples as part of BRAFi and BRAFi/MEKi clinical trials; however, none were willing to share their samples with us. Thus, we were limited to using RNAseq/microarray data even though this was not our preference since ABL1/2 mRNA upregulation does not always reflect increased activity (see below). Despite this caveat, we still were able to observe a correlation between increased ABL1 mRNA levels and resistance.

4. “Are the authors suggesting that ABL1 is dysregulated but not ABL2, as they present only the mRNA for ABL1 (Figure 1d)?” As indicated above, ABL1/2 mRNA upregulation (and increased protein expression), although required for ABL1/2 activation, is not sufficient to activate ABL1/2, and other events are also required such as loss of inhibitor binding, and tyrosine phosphorylation by tyrosine kinases²⁴. Thus, we might not necessarily expect to find increased mRNA levels in resistant samples, unless the other activating events also are present. However, we did, indeed, find that ABL1 (but not ABL2) mRNA is increased in resistant samples, indicating that regulation of ABL1 and ABL2 are distinct, which is not surprising given that the two proteins have different subcellular localizations (ABL1: nucleus/cytoplasm; ABL2: cytoplasm only), and thus, are regulated by similar but also unique mechanisms.

5. “The data presented in Figure 2 seems to suggest that nilotinib has similar effects in BRAF-sensitive and BRAF-resistant clones (Figure 2a, Figure 2c, Figure 2e), which do not fit with the model suggested by the authors that ABL is activated upon the development of BRAFi resistance.” We do not believe that the data in Fig. 2 are at odds with our hypothesis. ABL1/2 are activated in 40-60% of melanoma cell lines and patient samples.^{12,24} Indeed, ABL1/2 are highly activated in M14 parental cells;^{12,24} however, ABL1/2 activities are further potentiated following resistance (polyclonal populations were isolated). Data in Fig. 2 indicate that ABL1/2 have a role in intrinsic resistance (effect on parental cells) as well as on acquired resistance (subject of this manuscript). We focused the manuscript on acquired resistance since this is a pressing clinical problem. However, the fact that nilotinib also reverses intrinsic resistance does not decrease the clinical impact of our studies. In fact, the ability of nilotinib to dramatically prevent resistance, *in vivo*, may in part, be due to its ability to reverse intrinsic resistance.

These results provide further evidence that nilotinib is likely to work well not only in treatment-refractory patients but also as first-line therapy in treatment-naïve patients. These points were included in the results/discussion.

In conclusion, we thank the reviewers again for their insightful comments, which have led to experiments that we believe have strengthened the manuscript. We hope that the substantial changes we have made alleviate reviewer concerns such that our manuscript is now acceptable for publication in "*Nature Communications*".

References:

1. Yoshida K, Yamaguchi T, Natsume T, Kufe D, Miki Y. JNK phosphorylation of 14-3-3 proteins regulates nuclear targeting of c-Abl in the apoptotic response to DNA damage. *Nat Cell Biol* **7**, 278-285 (2005).
2. Nihira K, Taira N, Miki Y, Yoshida K. TTK/Mps1 controls nuclear targeting of c-Abl by 14-3-3-coupled phosphorylation in response to oxidative stress. *Oncogene* **27**, 7285-7295 (2008).
3. Plattner R, Koleske AJ, Kazlauskas A, Pendergast AM. Bidirectional Signaling Links the Abelson Kinases to the Platelet-Derived Growth Factor Receptor. *Mol Cell Biol* **24**, 2573-2583 (2004).
4. Finn AJ, Feng G, Pendergast AM. Postsynaptic requirement for Abl kinases in assembly of the neuromuscular junction. *Nature Neuroscience* **6**, 717-723 (2003).
5. Blake RA, *et al.* SU6656, a selective Src family kinase inhibitor, used to probe growth factor signaling. *Mol Cell Biol* **20**, 9018-9027 (2000).
6. Plattner R, Kadlec L, DeMali KA, Kazlauskas A, Pendergast AM. c-Abl is activated by growth factors and Src family kinases and has a role in the cellular response to PDGF. *Genes Dev* **13**, 2400-2411 (1999).
7. Wylie AA, *et al.* The allosteric inhibitor ABL001 enables dual targeting of BCR-ABL1. *Nature* **543**, 733-737 (2017).
8. Sonoshita M, *et al.* Promotion of colorectal cancer invasion and metastasis through activation of NOTCH-DAB1-ABL-RHOGEF protein TRIO. *Cancer Discov* **5**, 198-211 (2015).
9. Lin J, *et al.* Oncogenic activation of c-Abl in non-small cell lung cancer cells lacking FUS1 expression: inhibition of c-Abl by the tumor suppressor gene product Fus1. *Oncogene* **26**, 6989-6996 (2007).
10. Gu JJ, Rouse C, Xu X, Wang J, Onaitis MW, Pendergast AM. Inactivation of ABL kinases suppresses non-small cell lung cancer metastasis. *JCI Insight* **1**, e89647 (2016).
11. Testoni E, *et al.* Somatically mutated ABL1 is an actionable and essential NSCLC survival gene. *EMBO Mol Med* **8**, 105-116 (2016).
12. Jain A, Tripathi R, Turpin CP, Wang C, Plattner R. Abl kinase regulation by BRAF/ERK and cooperation with Akt in melanoma. *Oncogene* **36**, 4585-4596 (2017).
13. Fiore LS, *et al.* c-Abl and Arg induce cathepsin-mediated lysosomal degradation of the NM23-H1 metastasis suppressor in invasive cancer. *Oncogene* **33**, 4508-4520 (2014).

14. Srinivasan D, Plattner R. Activation of Abl tyrosine kinases promotes invasion of aggressive breast cancer cells. *Cancer Res* **66**, 5648-5655 (2006).
15. Ganguly SS, *et al.* c-Abl and Arg are activated in human primary melanomas, promote melanoma cell invasion via distinct pathways, and drive metastatic progression. *Oncogene* **31**, 1804-1816 (2012).
16. Gil-Henn H, Patsialou A, Wang Y, Warren MS, Condeelis JS, Koleske AJ. Arg/Abl2 promotes invasion and attenuates proliferation of breast cancer in vivo. *Oncogene* **32**, 2622-2630 (2013).
17. Chevalier C, *et al.* ABL tyrosine kinase inhibition variable effects on the invasive properties of different triple negative breast cancer cell lines. *PLoS One* **10**, e0118854 (2015).
18. Khatri A, Gu JJ, McKernan CM, Xu X, Pendergast AM. ABL kinase inhibition sensitizes primary lung adenocarcinomas to chemotherapy by promoting tumor cell differentiation. *Oncotarget* **10**, 1874-1886 (2019).
19. Khatri A, Kraft BD, Tata PR, Randell SH, Piantadosi CA, Pendergast AM. ABL kinase inhibition promotes lung regeneration through expansion of an SCGB1A1+ SPC+ cell population following bacterial pneumonia. *Proc Natl Acad Sci U S A* **116**, 1603-1612 (2019).
20. Meirson T, *et al.* Targeting invadopodia-mediated breast cancer metastasis by using ABL kinase inhibitors. *Oncotarget* **9**, 22158-22183 (2018).
21. Manley PWaZ, J. Drug Research leading to imatinib and beyond to nilotinib. In: *Polypharmacology in Drug Discovery* (ed Peters J-U). John Wiley & Sons, Inc (2012).
22. Kantarjian H, *et al.* Nilotinib in imatinib-resistant CML and Philadelphia chromosome-positive ALL. *N Engl J Med* **354**, 2542-2551 (2006).
23. Weisberg E, *et al.* Beneficial effects of combining nilotinib and imatinib in preclinical models of BCR-ABL+ leukemias. *Blood* **109**, 2112-2120 (2007).
24. Tripathi R, *et al.* Abl and Arg mediate cysteine cathepsin secretion to facilitate melanoma invasion and metastasis. *Sci Signal* **11**, (2018).
25. Li R, Pendergast AM. Arg kinase regulates epithelial cell polarity by targeting beta1-integrin and small GTPase pathways. *Curr Biol* **21**, 1534-1542 (2011).
26. Li X, *et al.* c-Abl and Arg tyrosine kinases regulate lysosomal degradation of the oncoprotein Galectin-3. *Cell Death Differ* **17**, 1277-1287 (2010).
27. Patel AB, O'Hare T, Deininger MW. Mechanisms of Resistance to ABL Kinase Inhibition in Chronic Myeloid Leukemia and the Development of Next Generation ABL Kinase Inhibitors. *Hematol Oncol Clin North Am* **31**, 589-612 (2017).
28. Tripathi R, Liu, Z, Plattner, R. Enabling Tumor Growth and Progression: Recent Progress in Unraveling the Functions of ABL kinases in Solid Tumor Cells. *Current Pharmacology Reports*, 1-13 (2018).

REVIEWER COMMENTS

Reviewer #2 (Remarks to the Author):

The authors have addressed all my points raised during the initial round of review by providing additional experimental data and making changes to the manuscript. No further comments.
Oliver Hantschel

Reviewer #3 (Remarks to the Author):

Despite the reviewers' otherwise comprehensive response, I remain very concerned about the specificity of the effects noted because of two points I raised in the initial review:

1. The specificity of the shRNA, which is the foundation for many of the genetic experiments shown. In the new submission, the authors show Figure S3d, which includes some growth assays and a Western blot. The Western blot was not correctly done. Specifically, where is the shScr/no cDNA lane (which is present in the growth experiments on the left)? Also, why is shScr used as the control for the cDNA?
2. The specificity of the pharmacologic effects, another key foundation for the experiments shown. The authors provide figure S2g, but the comparisons made are odd. Is there any difference between the ABL1/2-T315I cells treated with PLX or PLX+nilo? It appears that the nilo effects are significantly independent of ABL kinase inhibition.

Reviewer #4 (Remarks to the Author):

The authors have done a great job of addressing the comments. Figure 7 in particular has been added and helps to address outstanding questions.

Responses to Reviewer #3.

We thank all reviewers and the editors for taking the time to re-review our manuscript for "Nature Communications", particularly during this challenging time. Reviewers #2 and #4 were satisfied with the substantial additions we made to the manuscript and had no additional remarks. Reviewer #3 had two comments, which we responded to in-full below and in the relevant text (highlighted in yellow).

1. The specificity of the shRNA, which is the foundation for many of the genetic experiments shown. In the new submission, the authors show Figure S3d, which includes some growth assays and a Western blot. The Western blot was not correctly done. Specifically, where is the shScr/no cDNA lane (which is present in the growth experiments on the left)? Also, why is shScr used as the control for the cDNA? We thank the reviewer for noticing the mistake in the labeling of the western blot. The first lane was labeled incorrectly and should have been labeled "Vec". The figure is now corrected and shows the expression of ABL1 and ABL2 relative to the vector control in shABL1/2-transfected cells. The level of knockdown in shABL1/2 cells (vs. shScr) was shown in other figures (e.g. Fig. 3g). The labeling was clarified in the figure legend.

2. The specificity of the pharmacologic effects, another key foundation for the experiments shown. The authors provide figure S2g, but the comparisons made are odd. Is there any difference between the ABL1/2-T315I cells treated with PLX or PLX+nilo? It appears that the nilo effects are significantly independent of ABL kinase inhibition. In this experiment, mutant forms of ABL1/2 that cannot bind nilotinib (T315I) were expressed into M14-BR cells, and cell viability measured in the presence of nilotinib (PLX+nilotinib). If the target of nilotinib is not ABL1 and/or ABL2 (e.g. DDR), nilotinib would be predicted to be just as effective in vector vs. ABL1/2-T315I-transfected cells as nilotinib would retain the ability to bind/inhibit the other target (e.g. DDR). However, if ABL1/2 are the nilotinib targets, expression of nilotinib-resistant forms of ABL1/2 would be predicted to prevent nilotinib from reducing viability. This is why the comparison of interest is between vector and ABL1/2-T315I in the presence of nilotinib (PLX+nilotinib). Indeed, we found that expression of ABL1/2-T315I increases viability (compared to vector) in the presence of nilotinib. Thus, ABL1/2-T315I rescues nilotinib's ability to reduce survival indicating that nilotinib's effects are ABL1/2-dependent. This was clarified in the text (see Supplementary Figure Legend). The reviewer appears to be concerned that expression of ABL1/2-T315I increases viability in the absence of nilotinib (PLX alone). However, this result is expected as increased ABL1/2 expression (and thus, activity) would be predicted to increase viability/proliferation in resistant cells. Indeed, increased ABL1/2 expression promotes colony formation (Fig. 2h) and silencing ABL1/2 reduces viability (Fig. 3g). The fact that ABL1/2-T315I increases viability in the absence of nilotinib does not negate the fact that it rescues the effects of nilotinib, which would NOT be predicted to be the case if the nilotinib targets were NOT ABL1/2.